# DeepTravel: An End-to-End Agentic Reinforcement Learning Framework for Autonomous Travel Planning Agents

## Abstract

Travel planning (TP) agent has recently worked as an emerging building block to interact with external tools/resources for travel itinerary generation, ensuring enjoyable user experience. Despite its benefits, existing studies rely on hand-craft prompt and fixed agent workflow, hindering more flexible and autonomous TP agent. This paper proposes **DeepTravel**, an end-to-end agentic reinforcement learning framework for building autonomous travel planning agent, capable of autonomously planning, executing tools, and reflecting on tool responses to explore, verify, and refine intermediate actions in multi-step reasoning. To achieve this, we first construct a robust sandbox environment by caching transportation, accommodation and POI data, facilitating TP agent training without being constrained by real-world APIs limitations (e.g., inconsistent outputs). Moreover, we develop a hierarchical reward modeling system, where a trajectory-level verifier first checks spatiotemporal feasibility and filters unsatisfied travel itinerary, and then the turn-level verifier further validate itinerary's detail consistency with tool responses, enabling efficient and precise reward service. Finally, we propose the reply-augmented reinforcement learning method that enables TP agent to periodically replay from a failures experience buffer, emerging notable agentic capacity. We deploy trained TP agent on DiDi Enterprise Solutions App and conduct comprehensive online and offline evaluations, demonstrating that DeepTravel enables small-size LLMs (e.g., Qwen3-32B) to significantly outperform existing frontier LLMs such as OpenAI-o1/o3 and DeepSeek-R1 in travel planning tasks.

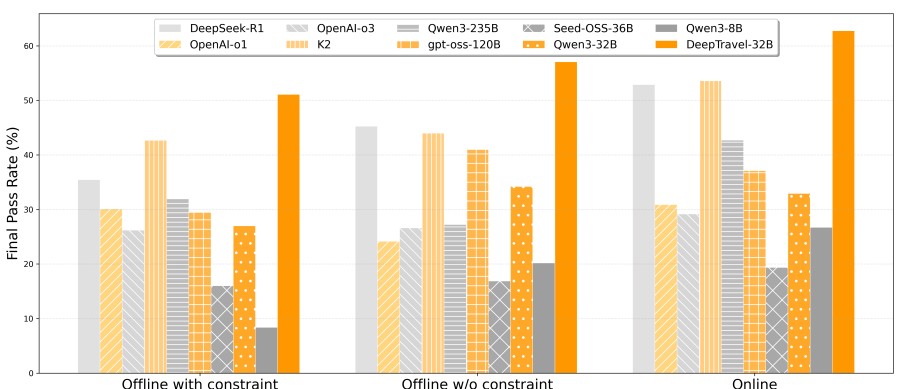

Figure 1: Performance of DeepTravel on synthesized offline benchmark and online user data.

## 1 Introduction

Travel planning (TP) aims to create a feasible itinerary Nguyen et al. (2023) that aligns with user preference by integrating multiple resources, such as accommodations, transportation, and Points-of-Interests (POIs). Recently, with the advances in natural language processing, large language models (LLMs) are widely used to build TP agents Chen et al. (2024), capable of invoking external

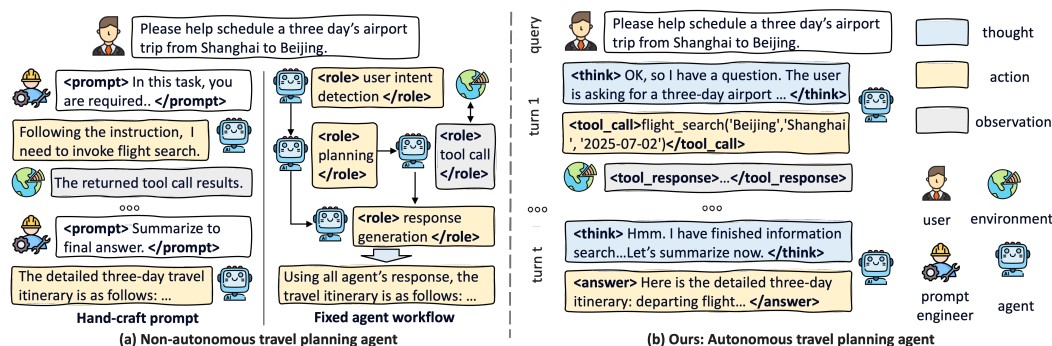

Figure 2: Comparison between existing studies and our autonomous travel planning agent paradigm.

tools/resources Gou et al. (2023) to generate travel itinerary, offering seamless experience in human mobility Tang et al. (2024). TP agent has gradually become a popular tool for the modern citizens.

In recent literature, many efforts have been devoted to construct TP agent. Most existing approaches primarily rely on carefully designed prompts, as illustrated in Figure 2(a). For example, TravelPlanner Xie et al. (2024) and TripTailor Wang et al. (2025) employ task-specific prompts to guide LLMs for tool invocation and itinerary generation. More recently, researchers have begun to integrate these prompt-engineering strategies into fixed agent workflow. For instance, PTS Shao et al. (2025) and RETAIL Deng et al. (2025) propose well-structured agent pipelines that enhance user intention understanding, enable effective tool interactions, and support accurate travel itinerary generation. However, these methods are *labor-intensive* and face challenges in *adapting to new user query or recovering from tool call failures*, limits more flexible and autonomous TP agents.

Agentic reinforcement learning (RL) Singh et al. (2025) has recently emerged and shown possibilities for building autonomous AI agent Jaech et al. (2024) by enabling agent to interact with tools in a dedicated environment and refine its reasoning based on the feedback it receives. For example, ReTool Feng et al. (2025), Kimi-Researcher MoonshotAI (2025) and WebSailor Li et al. (2025) utilize end-to-end agentic RL training to build the autonomous agent for math, deep research and web domain, respectively. These studies motivate us to propose a tailored agentic RL framework for autonomous TP agent construction, addressing the limitation in existing travel planning studies.

However, building an autonomous TP agent shown in Figure 2(b), that can progressively tackle complex TP tasks by autonomously interleaving tool calls and tool responses within the multi-turn reasoning, is non-trivial due to two key factors. *(1) Dynamic Travel Environment.* TP agents operate in a highly dynamic environment where information—such as hotel availability, pricing, and transportation options—changes continuously in real time. As a result, identical queries may produce inconsistent outputs over time due to updates in accommodations, transportation, and POI data sources. Training TP agents in such a constantly evolving real-world environment remains a significant challenge. *(2) Open-Ended Travel Task.* Unlike existing reinforcement learning with verified rewards (RLVR) paradigm Guo et al. (2025) on math or web domain, travel planning is an inherently open-ended task without explicit ground truth. For example, the generated travel itinerary may vary depending on personalized user preference and budget, making outcomes difficult to verify. How to construct reliable and scalable reward signals is challenging.

To address the aforementioned challenges, we propose **DeepTravel**, an end-to-end agentic RL training framework for autonomous travel planning agent construction. Specifically, we first construct a *Robust SandBox* by caching transportation, accommodation, and POI data from multiple real-world APIs across different timestamps, thereby simulating dynamic tool interactions. Within this sandbox, the TP agent can perform large-scale repeated trial-and-error learning while overcoming QPS limits and output inconsistencies. Moreover, we propose a *Hierarchical Reward Modeling* system, where a trajectory-level verifier ensures the spatiotemporal feasibility of generated itineraries and a turn-level verifier enforces fine-grained consistency with tool response, thereby yielding more efficient and reliable reward signals for training. Finally, we propose a *Reply-Augmented Reinforcement Learning* method to incentivize agentic reasoning capacity through sequential cold-start and RL process. Based on periodically replay from a failures experience buffer, the TP agent can learn and refine its previous reasoning actions, gradually emerging agentic travel planning capacity.

We deploy constructed TP agent in DiDi Enterprise Solutions App, and conduct evaluation using both of collected online real-world user data and offline synthetic data across varying task complexity. The experimental results demonstrate that DeepTravel enables small-size LLM backbones (e.g., Qwen3-32B) to outperform current state-of-the-art reasoning LLMs (e.g., OpenAI-o1/o3 and DeepSeek-R1) and prevailing RL algorithms (e.g., GRPO and DAPO).

Our contributions are summarized as follows: (1) We establish and deploy the first autonomous travel planning agent, offering new paradigm to advance existing TP studies. (2) We propose the first end-to-end agentic RL framework tailored to travel domain, which allows the training of TP agent under a robust sandbox environment, reliable reward service and periodical experience replay strategy. (3) Extensive experiments on online and offline data validate the effectiveness of proposed framework and uncover its exceptional performance across traval planning tasks.

## 2 PRELIMINARY

**Definition 1** *User Query. The user query $q$ is expressed in natural language, which indicates user's spatiotemporal travel intention and personalized preference. For example, a query "Please help schedule a three day's airport trip from Shanghai to Beijing" represents that the user wants to travel to Beijing by air and stay there for a duration of three days.*

**Definition 2** *Travel Itinerary. A travel itinerary $I$ is defined as a structured plan including accommodation, transportation, and detailed daily plan that integrates such as travel activity suggestions, exploration strategies for Points of Interest (POIs), and etc.*

**Problem 1** *Agentic Travel Planning. Given a query $q$, the travel agent generates travel itinerary $I$ to satisfy the trip requirements through automatically planing, executing tools, and reflecting on tool responses to explore, verify and refine intermediate actions in multi-turn reasoning process:*

$$\{\tau_t, a_t\} = \pi_\theta \left( q, \{\tau_1, a_1, o_1, \tau_2, a_2, o_2, \ldots, \tau_{t-1}, a_{t-1}, o_{t-1}\} \right), \tag{1}$$

*where $\pi_\theta$ is the policy of travel agent, and $\tau_{t-1}, a_{t-1}, o_{t-1}$ represent agent's thought, action, and observation from the environment in the $t-1$ turn, respectively. The generated travel itinerary $I$ is involved in the agent's action $a_t$ in the last turn.*

An agentic travel planning example is illustrated in Figure 2(b), where the autonomous TP agent carefully think (i.e., thought wrapped with `<think>` and `</think>`) before using external tools (i.e., action enclosed with `<tool_call>` and `</tool_call>`) and reflect on tool response (i.e., observation wrapped with `<tool_response>` and `</tool_response>`) to explore, verify and refine intermediate step in multi-turn reasoning process for generating travel itinerary.

## 3 DEEPTRAVEL

### 3.1 OVERVIEW

Figure 3 illustrates the overall pipeline of DeepTravel. *(1) Robust SandBox Construction* invovles toolkit annotation, mock data collection and update mechanism, thereby enabling simulated real-world tool interactions. *(2) Hierarchical Reward Modeling* proposes both of trajectory-level and turn-level verifier, which jointly provides reliable and efficient reward signal. *(3) Reply-Augmented Reinforcement Learning* first conducts SFT for agentic travel planning format cold start, then conducts RL with experience replay to further incentivize the agentic capacity of LLMs.

### 3.2 ROBUST SANDBOX CONSTRUCTION

The sandbox Lin et al. (2023) is served as a stable environment for TP agent to interact with tools, simulating real-world interaction while overcoming practical output inconsistency and API limits. We begin with toolkit annotation, then introduce the sandbox data collection and update mechanism.

#### 3.2.1 TOOLKIT ANNOTATION

We annotate six types of tools commonly used in travel planning. Table 1 summarizes the specifications of constructed toolkit, with parameter schemas and illustrative examples.

Table 1: The specifications of toolkit in sandbox.

| Type | Tool name | Tool call format | Tool response description |
|---|---|---|---|
| Transportation | flight search
train search
route planning | flight_search(depart_city_name, arrival_city_name, depart_date)
train_search(depart_city_name, arrival_city_name, depart_date)
route_planning(origin_name, destination_name, city_name) | feasible flight options
feasible train options
route, distance and time |
| Accommodation | hotel search | hotel_search(city_name,hotel_name,checkin_date,checkout_date) | available hotel condidate |
| Attraction | POI search
web search | poi_search(query,city_name)
web_search(query) | detailed address of POI
web page related to the query |

**Flight Search.** Flight search offers information about air transportation, a fundamental aspect of travel planning Shao et al. (2024a). In this work, we adopt the interface format from DiDi Enterprise Solutions (DiDi ES) App for this tool. Each tool call requires a departure city, an arrival city, and a departure date. For example, *"flight_search('Beijing', 'Shanghai', '2025-07-02')"* queries for flight options from *Beijing* to *Shanghai* on July 2, 2025.

**Train Search.** Similar to flight search, train search provides essential rail transportation information. We follow the DiDi ES interface format to build this tool. As shown in Table 1, each query includes a departure city, an arrival city, and a departure date.

**Route Planning.** Route planning is crucial for optimizing travel time and cost Fang et al. (2024). We leverage the route planning services provided by DiDi Map. Each tool call requires an origin name, a destination name, and a city name. For instance, *"route_planning('National Palace Museum', 'The Great Wall', 'Beijing')"* plans the route and calculates the distance/time details from the *National Palace Museum* to *The Great Wall* in Beijing.

**Hotel Search.** Hotel search enables the TP agent to find suitable accommodations based on user preferences Yang et al. (2025b). We also follow the DiDi ES interface format for this tool. Each tool call includes a city name, a hotel name, a check-in date, and a check-out date. For example, *"hotel_search('Beijing', 'Atour', '2025-07-02', '2025-07-05')"* searches for available rooms at the Atour hotel in Beijing from July 2 to July 5, 2025.

**POI Search.** POI search provides urban contextual semantics, which has been widely adopted in travel planning Xie et al. (2024). Similar with route planning, we directly utilize the POI search service provided by DiDi Map. Each tool call contains a query and a city name. For example, *"poi_search('The Great Wall', 'Beijing')"* helps obtain the geographic address of *The Great Wall*.

**Web Search.** The web contains a wealth of travel plans, serving as a valuable resource for generating itineraries Ni et al. (2025). We subscribe to web search services provided by Bocha AI. The tool call parameter is any query related to travel planning (e.g., *"web_search('Introduction to Beijing')"*).

### 3.2.2 MOCK DATA COLLECTION AND UPDATE MECHANISM

To simulate the dynamic nature of real-world tool interactions, where prices and availability of hotels, trains, and flights fluctuate over time, we propose a data caching mechanism. This mechanism addresses the challenge of real-world API QPS limits and inconsistent information retrieval during each search, which can hinder the agent's ability to learn from previously failed cases.

Specifically, we maintain a daily-refreshed database that incorporates an on-demand caching strategy for flight, train, and hotel search data. When a new query is received, the system automatically stores the latest records into the database. This mechanism allows the TP agent to re-access earlier tool response during RL training, enabling it to repeatedly learn from unsuccessful cases through exploring, verifying and refining its intermediate reasoning trajectories Shang et al. (2025).

### 3.3 HIERARCHICAL REWARD MODELING

Then, we present our reward modeling system. This system comprises a trajectory-level verifier and a turn-level verifier, designed to provide efficient and reliable reward signals for agent training.

**Trajectory-Level Verifier.** This verifier assesses the overall spatiotemporal feasibility of the generated travel itinerary. Given a complete reasoning trajectory $\{\tau_1, a_1, o_1, \tau_2, a_2, o_2, \ldots, \tau_t, a_t\}$, trajectory-level verifier checks whether the final travel itinerary $a_t$ adheres to essential spatiotemporal constraints Chaudhuri et al. (2025). These constraints include such as logical sequence of events, geographic plausibility, and satisfaction of user requirements (e.g., visiting specific POIs within a

Figure 3: An overview of DeepTravel.

given timeframe). This coarse-grained evaluation efficiently filters out invalid itineraries, ensuring that only potentially valid plans proceed to the next level of verification.

**Turn-Level Verifier.** Upon successful verification by the trajectory-level verifier, the turn-level verifier performs a more granular evaluation. This verifier examines the consistency between the agent's final travel itinerary $a_t$ and the information obtained by the external tools at each turn Zeng et al. (2025). Specifically, for each turn $i$ from 1 to $t-1$, the verifier assesses whether $a_t$ accurately reflect the tool response $o_i$. By systematically verifying each turn, it helps identify factual hallucination/mistakes of LLM-generated travel itinerary.

**Joint Reward Reweighting.** Two verifier are combined using a joint reward reweighting strategy to provide reward signal. If the trajectory-level verifier detects a violation, the final reward $r$ is immediately set to 0, saving computational resources. If the trajectory passes the trajectory-level verification, the turn-level verifier assesses each turn. The final reward $r$ is set to 1 only if every turn passes verification, indicating a fully consistent and valid travel itinerary.

This hierarchical structure ensures both the efficiency and reliability of the reward modeling system. In practical implementation, we build many travel-oriented rubrics Huang et al. (2025) for trajectory-level and turn-level verifier, respectively. Based on human generated rubrics, we prompt DeepSeek-R1 based verifier to provide reward modeling service. Details could be found in Appendix A.2.

### 3.4 REPLAY-AUGMENTED REINFORCEMENT LEARNING

This section details relay-augmented reinforcement learning, which is a two-stage process. We first employ SFT to initialize reasoning format of TP agent. Then, we leverage RL to further enhance agent's reasoning capacity, and enable it to periodically learn from previous failed experience.

#### 3.4.1 REASONING FORMAT COLD START WITH SUPERVISED FINE-TUNING

**Cold-Start Data Synthesis and Filtering.** We distill multi-turn trajectories from DeepSeek-R1 under the sandbox, yielding complete traces $y = \{\tau_1, a_1, o_1, \ldots, \tau_t, a_t\}$ that interleave thoughts, tool calls, tool responses, and final answer. Thoughts $\tau_i$ are wrapped by `<think>...</think>`, actions $a_i$ are either function calls enclosed by `<tool_call>...</tool_call>` or the final itinerary answer $a_t$ enclosed by `<answer>...</answer>`, and observations $o_i$ are tool responses enclosed by `<tool_response>...</tool_response>`. Then, we utilize the constructed reward modeling system to filter incorrect trajectory, and finally we apply strict format checks to retain only sequences correctly segmented by the special tags.

**Training Objective.** We train the TP agent to follow a system prompt $T$ and to reproduce verified tool-integrated trajectories. The instruction input concatenates $T$ with the user query $q$, and the output is the verified trajectory $y$. In practical training process, the tokens corresponding to the agent's environmental observations $o_i$ are masked out from the loss calculation Jin et al. (2025). The detailed prompting template could be found in Appendix A.2.

#### 3.4.2 REINFORCEMENT LEARNING WITH EXPERIENCE REPLAY

After cold-start, we derive a two-phase process that first saves verified unsuccessful trajectories as a query buffer and then replays them in subsequent training steps Zhang et al. (2025b).

**Rollout and Replay Strategy.** Following the sampling procedure in Group Relative Policy Optimization (GRPO) Shao et al. (2024b), we sample a group of trajectories $\{y_1, y_2, .., y_n\}$ for each

query $q$. If none of the trajectories in the group yields a verified correct answer, we store the query in an experience buffer $B$ for later replay. The motivation is that, after subsequent RL training steps, the improved policy may generalize to handle previous failed hard sample Xie et al. (2025).

**Policy Optimization.** Set RL training dataset as $D$, experience buffer as $B$, which is replayed after fixed training step $\gamma$. We formulate the optimization goal as follows:

$$
\max_{\pi_\theta} \mathbb{E}_{q \sim \{D, B\}, \{y_i\}_{i=1}^n \sim \pi(y|q)} \left[ \frac{1}{n} \sum_{i=1}^n \Big( \min \Big( \frac{\pi_\theta(y_i \mid q)}{\pi_{ref}(y_i \mid q)} A_i, \right.
$$
$$
\left. \text{clip}\Big( \frac{\pi_\theta(y_i \mid q)}{\pi_{ref}(y_i \mid q)}, 1 - \varepsilon, 1 + \varepsilon \Big) A_i \Big) \Big) - \beta \mathbb{D}_{\text{KL}} \left[ \pi_\theta \| \pi_{\text{ref}} \right] \right]
\tag{2}
$$

where $\varepsilon$, $\beta$ are hyperparameters, $n$ is the rollout size, $\mathbb{D}_{\text{KL}}$ denotes the KL-divergence, and $A_i = r_i - avg(r)/std(r)$ represents the advantage, which is computed based on the group rewards $r = \{r_1, r_2, .., r_n\}$. In this work, we propose to filter out samples when the standard deviation of group rewards satisfies $std(r) \leq \eta$, where $\eta$ is set to 0.1. This strategy aims to exclude samples that are either too simple or too hard, where the agent receives similar rewards even under large-size rollouts, thereby encouraging more effective exploration of the current policy. In addition, we utilize loss masking operation for tool responses tokens (wrapped with `<tool_response>` and `</tool_response>`) to ensure policy gradient is computed only over agent-generated tokens.

## 4 EXPERIMENTS

### 4.1 EXPERIMENTAL SETUP

**Data Curation.** Our method is evaluated on four benchmarks, constructed using a combination of real-world online user data from DiDi ES and synthetic offline data: (1) **Online data**: we collected real-world user queries from DiDi ES App between June 1st and August 31st, spanning three months and totaling 6,224 queries. (2) **Offline data**: we synthesized 500 user queries under two distinct settings respectively: a) queries **without constraints**, such as travel budget, or personalized preferences; (b) queries generated **with constraints**. We annotated query complexity in both settings, categorizing each query as **easy, medium, or complex**. Further data curation details and dataset statistic information are provided in Appendix A.3.

**Baselines.** We compare our method to nine reasoning LLMs, each is derived as a TP agent under the same framework. These baselines include DeepSeek-R1 Guo et al. (2025), OpenAI-o1 Jaech et al. (2024), OpenAI-o3, K2 Team et al. (2025), Qwen3-235B, gpt-oss-120B OpenAI (2025), Seed-OSS-36B Team (2025), Qwen3-32B, and Qwen3-8B Yang et al. (2025a). In addition, we also compare our proposed DeepTravel framework with three representative RL training methods, i.e., PPO Schulman et al. (2017), GRPO Shao et al. (2024b) and DAPO Yu et al. (2025).

**Metric and Evaluation Protocol.** We use the Final Pass Rate from TravelPlanner Xie et al. (2024) as our evaluation metric. For the evaluation protocol, we apply our constructed reward modeling system to estimate the final pass rate. Additionally, we randomly sample 50 cases from both online and offline results and have human annotators assess whether the generated travel itineraries pass.

**Training Details.** We leverage the Qwen3-8B and Qwen3-32B to perform SFT and RL, resulting in the DeepTravel-8B and DeepTravel-32B. For the SFT cold-start, we use 1K training samples, with a batch size of 32 and a learning rate of 5e-6, over 2 epochs. For RL training, we select 500 high-quality samples and employ the proposed Replay-Augmented RL algorithm. During RL, we set the rollout size to 8 and use a learning rate of 5e-7. Additionally, the maximum response length of agent is set as 32K tokens, and the maximum interaction turns is limited to 8. The training of Qwen3-8B and Qwen3-32B requires 8 and 32 H800 NVIDIA GPUs respectively, and the training process takes approximately 30 hours per hundreds steps. We provide more details in Appendix A.4

### 4.2 MAIN RESULTS

**Comparison with Existing Reasoning Agents.** We compare DeepTravel across the SFT cold-start stage and RL training process. As reported in Table 2. Overall, DeepTravel achieves significant improvement compared with the the state-of-the-art reasoning agents using both of online and offline

Table 2: Overall Final Pass Rate (%) results on both of synthesized offline travel planning benchmarks and real-world online user data on DiDi ES application. The best results are **bolded**, and the best baseline results in each setting are underlined.

| Model | Offline | | | | | | Online | Human Evaluation |
| | Without constraint | | | With constraint | | | | |
| | Easy | Medium | Hard | Easy | Medium | Hard | | |
|---|---|---|---|---|---|---|---|---|
| DeepSeek-R1 | 45.55 | 34.74 | 26.00 | 65.36 | 43.33 | 27.09 | 52.89 | 72.00 |
| OpenAI-o1 | 36.57 | 33.16 | 20.60 | 30.36 | 24.44 | 17.69 | 30.88 | 54.00 |
| OpenAI-o3 | 37.30 | 20.11 | 21.19 | 37.50 | 26.67 | 15.69 | 29.17 | 52.00 |
| K2 | 54.01 | 48.42 | 25.52 | 57.14 | 53.33 | 21.40 | 53.56 | 64.00 |
| Qwen3-235B | 38.69 | 36.84 | 20.24 | 44.64 | 26.67 | 10.37 | 42.70 | 52.00 |
| gpt-oss-120B | 40.15 | 27.37 | 20.83 | 64.29 | 42.22 | 16.39 | 37.11 | 48.00 |
| Seed-OSS-36B | 23.65 | 13.16 | 11.19 | 25.00 | 13.33 | 12.34 | 19.36 | 20.00 |
| Qwen3-32B | 29.85 | 27.89 | 23.21 | 53.57 | 25.00 | 9.03 | 32.94 | 38.00 |
| Qwen3-8B | 10.95 | 9.47 | 4.76 | 28.57 | 26.67 | 5.35 | 26.72 | 26.00 |
| DeepTravel-8B-Cold-Start | 41.09 | 31.58 | 12.64 | 56.07 | 28.89 | 12.37 | 40.00 | 58.00 |
| DeepTravel-8B-RL | 54.25 | 36.84 | 20.24 | 64.86 | 41.89 | 21.40 | 49.75 | 70.00 |
| DeepTravel-32B-Cold-Start | 56.42 | 32.95 | 25.60 | 61.07 | 40.44 | 17.52 | 50.03 | 66.00 |
| DeepTravel-32B-RL | **69.34** | **54.74** | **29.17** | **73.21** | **62.22** | **35.75** | **62.77** | **82.00** |

Table 3: Compatison of DeepTravel with existing RL alogrithms on Qwen3-8B.

| Model | Offline | | | | | | Online | Human Evaluation |
| | Without constraint | | | With constraint | | | | |
| | Easy | Medium | Hard | Easy | Medium | Hard | | |
|---|---|---|---|---|---|---|---|---|
| Base (Qwen3-8B) | 10.95 | 9.47 | 4.76 | 28.57 | 26.67 | 5.35 | 26.72 | 26.00 |
| with PPO | 48.26 | 33.25 | 14.62 | 60.05 | 34.86 | 15.04 | 45.63 | 62.00 |
| with GRPO | 52.36 | 34.06 | 13.52 | 61.78 | 36.65 | 15.82 | 47.78 | 64.00 |
| with DAPO | 52.06 | 35.52 | 15.04 | 62.24 | 40.02 | 16.54 | 46.07 | 64.00 |
| with DeepTravel | **54.25** | **36.84** | **20.24** | **64.86** | **41.89** | **21.40** | **49.75** | **70.00** |

evaluation setting. In addition, We highlight two key observations: **(i) DeepTravel substantially boosts small-size LLMs**. For instance, Qwen3-8/32B is improved to the state-of-the-art levels, matching and even surpassing more heavily and much larger frontier LLMs. On offline without constraint setting, DeepTravel-8B and DeepTravel-32B achieves an final pass rate of 54.25% and 69.34%, outperforming K2 by 0.1% and 28.9%, respectively. For other setting, DeepTravel-8B achieves comparable performance and DeepTravel-32B consistently outperforms frontier reasoning LLMs, such as DeepSeek-R1, OpenAI-o1 and OpenAI-o3. **(ii) The agentic RL training continually improves domain-specific reasoning capacity**. As reported, while cold-start stage could establish a strong initial policy compared to base model, the following agentic RL yields surprisingly performance improvement. Specifically, agentic RL further boost initial cold-start policy of DeepTravel-8B and DeepTravel-32B by 24% (40.00 to 49.75) and 25.5% (50.03 to 62.77) on online experimental setting, respectively. The improvement in the offline setting is also significant.

**Comparison with Existing RL Alogrithms.** We also compare DeepTrave with three recent public RL methods on DeepTravel-8B-Cold-Start backbone. For each RL methods, we run 100 training steps with the same training sample. Table 3 reports the results. We highlight the following observations: while all online RL methods improve the base model, our proposed DeepTravel significantly outperforms existing RL methods on hard problems, likely due to its reply mechanism.

## 4.3 ABLATION STUDY

To validate the effectiveness of each module in DeepTravel, we conduct an ablation study on the Qwen3-8B dataset. Specifically, we compare the following variants. (1) DeepTravel-8B w/o ER removes the **E**xperience **R**eplay module in RL training process. (2) DeepTravel-8B w/o CS removes the SFT-based **C**old **S**tart stage before conduct reinforcement learning. (3) DeepTravel-8B w/o Traj removes the **Traj**ectory-Level verifier in reinforcement learning training process. (4) DeepTravel-8B w/o Turn removes the **Turn**-Level verifier in reinforcement learning training process.

As shown in Table 4, we obtain the following observations. First, the experience replay strategy is important for the training. Removing it will decrease model performance. Second, the cold-start stage seems to be critical for RL training as we obtain significant performance decrease after

Table 4: Ablation study of cold-start and RL on Qwen3-8B.

| Model variants | Offline | | | | | | Online | Human Evaluation |
| | Without constraint | | | With constraint | | | | |
| | Easy | Medium | Hard | Easy | Medium | Hard | | |
| --- | --- | --- | --- | --- | --- | --- | --- | --- |
| DeepTravel-8B w/o ER | 51.01 | 32.21 | 8.81 | 60.86 | 35.00 | 8.75 | 40.00 | 66.00 |
| DeepTravel-8B w/o CS | 45.99 | 25.26 | 16.79 | 53.57 | 35.56 | 22.18 | 32.45 | 48.00 |
| DeepTravel-8B w/o Traj | 50.26 | 35.47 | 18.24 | 61.06 | 33.25 | 20.75 | 26.52 | 66.00 |
| DeepTravel-8B w/o Turn | 52.05 | 28.04 | 5.25 | 59.04 | 14.24 | 10.76 | 32.45 | 58.00 |
| DeepTravel-8B | 54.01 | 36.84 | 20.24 | 64.86 | 41.89 | 21.40 | 49.75 | 70.00 |

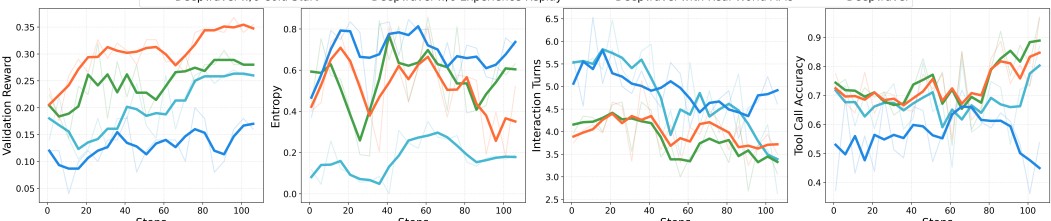

Figure 4: Validation reward (final pass rate), model entropy, average interaction turn and tool call accuracy (success rate) throughout agentic RL training process.

removing it. The potential reason lies on that the cold-start will help LLMs learn basic tool usage, instruction following capacity. Finally, we observe that the turn-level verifier contributes more to the model performance. When removing it, agent's performance decreases and it performs poorly on hard problem. The reason may lie that more complex problem requires verification turn-by-turn. However, the trajectory-level verifier also proves important, as its removal causes a performance decline as well. In addition to its contribution to accuracy, the trajectory-level verifier enhances training efficiency by removing the need of fine-grained turn-level verification.

## 4.4 IN-DEPTH ANALYSIS

**Agentic RL Training Analysis.** We present the validation reward, model entropy, average interaction turns, and tool-call accuracy throughout the RL training process in Figure 4. To evaluate the effectiveness of DeepTravel, we highlight the following findings: **(i) Impact of the cold-start and experience replay strategies.** The cold-start strategy helps initialize a reasonably effective policy, particularly in terms of producing a more appropriate number of tool-interaction turns (whereas the base model tends to overuse tool calls). The experience replay strategy contributes little in the very early training stages. However, by progressively replaying previously failed samples, it steadily enhances the model's capacity and eventually leads to substantial improvements over the no-replay baseline in later training steps. **(ii) Impact of the sandbox.** We compare agentic RL training with real-world APIs and with the constructed sandbox. Tool-call accuracy with real APIs is unstable and consistently lower than that achieved in the sandbox. Under these circumstances, the TP agent shows no clear reward improvement, highlighting the importance of a stable sandbox environment for agentic Rl training. **(iii) Non-decreasing entropy in agentic RL.** We further observe a non-decreasing entropy phenomenon during agentic RL training, which is also posed by several very recent studies Dong et al. (2025). We think the behind reason lies on that the TP agent need to continually adapt its policy to the dynamically changing responses of external tools.

**Real-World User Study.** We conducted a real-world user study based on evaluation dimensions defined by DiDi's ES product manager and annotation team, with the results summarized in Figure 5. Overall, both the cold-start and RL approaches improved user satisfaction across the seven evaluation dimensions. In particular, the cold-start method substantially enhanced the model's fundamental capabilities, especially in understanding user intentions and in improving the completeness, feasibility, and clarity of the generated travel itineraries. However, for more advanced capabilities—such as capturing and satisfying personalized preferences—the cold-start approach alone proved insufficient, suggesting that these aspects may require large-scale exploration during the RL stage. Finally, we observed that both the base model and the cold-start model suffered from severe hallucination issues, with factual error rates reaching up to 50%. RL training is able to effectively address this problem, reducing hallucinations to below 20%. More annotation insights is in Appendix A.5

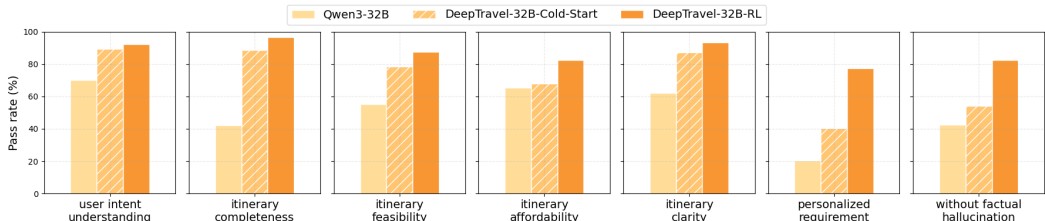

Figure 5: Capacity comparison of the autonomous TP agent across 7 human-annotated dimensions, evaluated on 50 randomly sampled real-world online user case.

## 5 RELATED WORK

### 5.1 LLMs AS TRAVEL PLANNING AGENT

LLMs have reshaped travel planning (TP) by enabling LLM-powered TP agent to interact with external tools for itinerary generation that aligns with user preferences. In the literature, two major paradigms have emerged to construct TP agent: (i) hand-craft prompt tuning, and (ii) fixed agent workflow design. Hand-craft prompt tuning approaches Shao et al. (2025)—including TravelPlanner Xie et al. (2024), TripTailor Wang et al. (2025), and ChinaTravel Shao et al. (2024a)—decompose the end-to-end task into multiple sub-steps Ni et al. (2025) and introduce tailored evaluation metrics for each stage. While effective, their practical value is limited by weak grounding to dynamic real-world environment (e.g., tool availability). To this end, recent work integrates prompt strategies into well-structured agent pipelines. Representative studies include TravelAgent Chen et al. (2024), PTS Shao et al. (2025) and RETAIL Deng et al. (2025) design fixed workflows to enhance intention understanding, orchestrate external tools, and ensure end-to-end itinerary generation. However, they are still labor-intensive to build and maintain, and they generalize poorly to new user queries or changing tools and resources, limiting the flexibility and autonomy of TP agents.

### 5.2 AGENTIC REINFORCEMENT LEARNING FOR LLMs

Agentic reinforcement learning (RL) has recently been widely applied across domains to build autonomous AI agents Zhang et al. (2025a), wherein the agent interacts with tools in a dedicated sandbox environment and iteratively improves its policy based on received reward feedback Shang et al. (2025). For instance, ReTool Feng et al. (2025), Kimi-Researcher MoonshotAI (2025) and WebSailor Li et al. (2025) are constructed reasoning agent in math, deep research and web domain. In addition, many recent work like rStar2-Agent Shang et al. (2025) and AgentGym-RL Xi et al. (2025) make attempts to propose a unified agentic RL training framework across diverse domains, facilitating the construction of foundation agent. Nevertheless, the application of agentic RL in travel planning domain remains unexplored.

## 6 CONCLUSION, LIMITATION AND FUTURE WORK

In this work, we propose DeepTravel, the first end-to-end agentic RL training framework to build autonomous travel planning (TP) agent, offering new paradigm for current TP studies. We first construct a robust sandbox, where the TP agent could be trained without limitation of real-world APIs issues (e.g., QPS limits and inconsistency outputs). Then, we propose a hierarchical reward modeling system, which first devise a coarse-grained trajectory verifier for high-level spatiotemporal requirement verification, and then use a turn-level verifier to verify agent's answer step-by-step. Finally, we propose a replay-augmented reinforcement learning alogrithm, which allow travel agent to periodically replay previous failed case, improve its out-of-domain generalization capacity. We deploy the resulted autonomous TP agent in DiDi ES App, demonstrating the deployment value of DeepTravel. Extensive experiments on online production environment and offline synthetic data show that DeepTravel enable small-size LLMs (e.g., Qwen3-8B/32B) to significantly outperform frontier reasoning LLMs, such as DeepSeek-R1 and OpenAI-o1/o3. However, DeepTravel relies on a carefully designed reward system, which limits its extensibility. In the future, we aim to develop a more flexible reward model and extend this framework to other domains.

## ETHICS AND REPRODUCIBILITY STATEMENT

**Ethics statement.** This work leverages real-world user data from the DiDi Enterprise Solution App for model training and evaluation. However, we do not store any personal information or release real user queries, so there are no additional privacy or fairness concerns. For synthesized user query, we provide query case example in Appendix A.6 to ease reader understanding. For human annotations used in model evaluation, we also provide the full set of evaluation dimensions in Appendix A.5 to facilitate replication of our annotation process.

**Reproducibility statement.** To ensure reproducibility, we provide a detailed description of the training prompts in Appendix A.2 and outline the data curation pipeline in Appendix A.3. We believe this information will help the research community reproduce our results. Because the DeepTravel includes proprietary interfaces to DiDi Enterprise Solution, we cannot release a unified sandbox and training implementation. Nevertheless, we provide comprehensive training details in Appendix A.4, including compute resources and monitoring metrics to aid reproductionn and understanding.

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

# DeepTravel: An End-to-End Agentic Reinforcement Learning Framework for Autonomous Travel Planning Agents
## *Supplementary Material*

## CONTENTS

## A    APPENDIX

### A.1    USAGE OF LARGE LANGUAGE MODELS

In this paper, we primarily use large language models (LLMs) to generate figure plots. We also employ LLMs to identify and correct clear grammatical errors in the authors' drafted paragraphs.

### A.2    PROMPT TEMPLATE

In this section, we provide prompt used in this work, including system prompt of DeepTravel and the reward model prompt template of the trajectory-level and turn-level verifier.

---

**A.2.2 Prompt for Trajectory-Level Verifier**

As a travel planning judger, you will evaluate whether the agent's response adheres to the following criteria.

You will receive:
1. [Query]: Contains the user's needs and travel constraints.
2. [Agent's Response]: The AI travel assistant's final response, which you will need to verify. Please strictly follow the following [Evaluation Rubrics] to make quality assessments.

**Evaluation Rubrics**
1. [Is the answer complete?]... 2. [Is the main requirement understood accurately?]... 3. [Is the logic sound?]... 4. [Are other constraints met?]... 5. [Are specific requirements met?]... 6. [Emergency backup plan?]...

**Available Tools**

```
poi_search(query, city_name, **kwargs)

route_planning(origin, destination, city_name)

flight_search(depart_city, arrival_city, depart_date, **kwargs)

train_search(depart_city, arrival_city, depart_date, **kwargs)

hotel_search(city_name, checkin_date, checkout_date, **kwargs)

web_search(query)
```

**Evaluation Output**
Evaluation Reason: Provide the analysis process and reasons.
Final Conclusion: Very satisfied or Very satisfied but did not address unexpected situations or Basically satisfied, other constraints or specific requirements were not met or Dissatisfied, logically unreasonable or Dissatisfied, main requirements misunderstood or Dissatisfied, incomplete answer

Let's get started! Return the evaluation reason and final conclusion.

---

**System Prompt of DeepTrabel.** We provide the system prompt and reward model prompt of our constructed travel planning agent, enable it to autonomously planning, executing tools and reflecting on tool responses in multi-step reasoning process for travel itinerary generation. As shown in Tabel A.2.1, we provide essential reasoning format, toolkit information and answer rule in the system prompt, guiding agent's behavior.

**Prompt Template of Trajectory-Level Verifier.** In this work, we construct the reward modeling system using hand-crafted evaluation rubrics. For the trajectory-level verifier, we provide the user query and the agent's final response. The primary objective is to assess whether the response adheres to the annotated, coarse-grained spatiotemporal principles—such as answer completeness, alignment with user intent, and conformance to the logical structure of travel planning. These foundational principles are operationalized into six evaluation rubrics, shown in Table A.2.2.

### A.2.1 System Prompt of DeepTravel

You are a business travel assistant named Xiao Di on the DiDi Enterprise Solution, jointly developed by DiDi Enterprise and DiDi. You are very intelligent and capable of uncovering users' latent needs to surprise them. Based on user input, you respond in different ways.

Follow this template:

```
<think>...</think>
<tool_call>...</tool_call>
<tool_response>...</tool_response>
<tool_response_thinking>...</tool_response_thinking>
...
<think_call_thinking>...</think_call_thinking>
<tool_call>...</tool_call>
<tool_response>...</tool_response>
<tool_response_thinking>...</tool_response_thinking>
...
<answer>Place the final result here.</answer>
```

**Toolkit Information**:

Flight search tool is a combined tool that integrates the functions of POI search and taxi search. When users need to connect large and small transportation, they can directly set depart_poi or arrival_poi. The usage method is as follows:

`flight_search(depart_city,arrival_city,depart_date,**kwargs)`

Train ticket search tool is a combined tool that can query direct and transfer information between two stations. The usage method is as follows:

`train_search(depart_city, arrival_city, depart_date, **kwargs)`

Hotel search tool is a function call, and the usage method is as follows:

`hotel_search(city_name,checkin_date,checkout_date,**kwargs)`

Web search is a function that allows you to search the internet for real-time information. The usage method is as follows:

`web_search(query)`

POI search is a function that allows you to query addresses based on location descriptions. The usage method is as follows:

`poi_search(query, city_name, **kwargs)`

Route planning is a function that allows you to obtain distance and time information between two locations within the same city. The usage method is as follows:

`route_planning(origin, destination, city_name)`

**Answer Rule**

The complete travel plan follows the format below (displayed in the order of departure - hotel - return).

Example:

Based on your request/ business trip form, we have planned a business trip from xx city to xx city from x month x to x month x. The specific arrangements are as follows: ...

Departure recommendations (use markdown third-level title, the default number of recommendations does not exceed 2, departure date) ...

Hotel recommendations (use markdown third-level title , the default number of recommendations does not exceed 2) ...

Return recommendations (use markdown third-level title , the default number of recommendations does not exceed 2, departure date) ...

OK. Let's Start!!

**Prompt Template of Turn-Level Verifier.** Beyond the trajectory-level verifier, we also employ a turn-level verifier that evaluates the agent's responses turn by turn. To achieve this, we incorporate additional tool-response information into the prompt. At each turn, we instruct the LLM to generate both the reasoning behind its evaluation and a final conclusion. As shown in Table A.2.3, the verifier is prompted to provide step-by-step evaluation reasoning along with a final judgment for every turn.

---

**A.2.3 Prompt for Turn-Level Verifier**

As a travel planning judger, you will evaluate whether the agent's response adheres to the following criteria.

You will receive:
1. [Query]: Contains the user's needs and travel constraints.
2. [Agent's Response]: The AI travel assistant's final response, which you will need to verify. Please strictly follow the following [Evaluation Rubrics] to make quality assessments.
3. [Tool response used for agent's response generation]: What information the AI assistant used for response generation:

`<tool_response>..</tool_response>`

**Evaluation Rubrics**
1.[Is the tool call parameters/logic correct?]... 2. [Is the agent's response accurately reflect the tool response?]...

**Available Tools**

`poi_search(query, city_name, **kwargs)`

`route_planning(origin, destination, city_name)`

`flight_search(depart_city, arrival_city, depart_date, **kwargs)`

`train_search(depart_city, arrival_city, depart_date, **kwargs)`

`hotel_search(city_name, checkin_date, checkout_date, **kwargs)`

`web_search(query)`

**Evaluation Output**
Evaluation Reason: Provide the analysis process and reasons.
Final Conclusion: Satisfied or Unsatisfied where inconsistent information between agent response and tool response, or Unsatisfied with tool call logic error.

Let's get started! Return the evaluation reason and final conclusion.

---

## A.3 DATA CURATION

Except for the prompt design for TP agent and reward model, one important part is the data curation. In this section, we first introduce the pipeline of data synthesis, and then introduce the detailed statistics information used in DeepTravel, including SFT Cold-start stage, RL training, offline testing and online testsing.

### A.3.1 DATA SYNTHESIZATION PIPELINE

In this paper, the primary goal of data synthesis is to construct user queries. Each user query is decomposed into a set of atomic intents, such as *origin*, *destination*, *departure time*, *arrival time*, *budget*, and so on. In our method, we systematically enumerate combinations of atomic intents and then inversely generate natural user queries that correspond to these intent structures.

Once the initial pool of queries is obtained, we perform repeated sampling with the Qwen-3-32B model. Each candidate query is then evaluated with our reward modeling system, which assigns a difficulty score. For SFT Cold-Start, we primarily use relatively simple queries, while retaining

a small proportion of difficult ones to improve model robustness. For RL, on the contrary, we emphasize complex queries, but still include a small fraction of simple cases to ensure coverage.

Before integrating queries into RL training, we additionally conduct a manual inspection stage. This human filtering step is crucial to remove semantically invalid or ill-posed queries. We found that the quality of queries significantly impacts RL training outcomes, and ensuring that queries are both challenging and learnable is essential for stable optimization.

For test data, we construct evaluation sets by distinguishing queries based on whether they involve explicit user personalization constraints (e.g., budget restrictions or individual travel preferences). Accordingly, we build two categories: *offline with constraint* and *offline without constraint*. We carefully select 500 queries for each category, resulting in balanced test datasets that cover both personalized and non-personalized travel scenarios.

### A.3.2 DATASET STATISTICS

In this section, we present the dataset statistics for SFT Cold-Start, RL Training, and testing.

**SFT Cold-Start.** For each model iteration and update, we employ the data synthesis pipeline to generate user queries. We then conduct repeated sampling using DeepSeek-R1. After filtering the samples with the constructed reward model, we obtain approximately 1K trajectories for the backbone cold-start.

**RL Training.** The training dataset consists of about 500 high-quality samples, all of which have been double-checked by human annotators. In the RL training process, we set the validation size to 50 and the training size to 450.

**Online and Offline Testing.** We use both of online and offline testing strategy to validate the model improvement of DeepTravel framework. For offline testing, We use 500 test samples with constraints (156 easy, 45 medium, and 299 hard samples) and 500 test samples without constraints (222 easy, 78 medium, and 200 hard samples). For online testing, we use 6,224 queries collected from the online production environment of the DiDi ES App.

### A.4 TRAINING DETAILS

For SFT-based cold-start, we use Megatron-LM for fine-tuning, and build our RL training framework on top of verl. In the RL training process, we select DeepSeek-R1 as the backbone of our verifier. For an 8B-parameter LLM, all training can be completed on a single node with 8 H800 GPUs. For a 32B-parameter LLM, training requires 32 H800 GPUs across 4 nodes. SFT takes about 2–3 hours for 1,000 training samples. RL takes about 30 hours per 100 steps for 500 samples.

Below we detail the supervision signals and diagnostics we track to ensure stable and effective training of DeepTravel. For the TP agent, we continuously monitor entropy, gradient norm, average response length, reward, and average turn count. We also track metrics tied to the broader RL loop—tool-call accuracy and external verifier success rate—to capture the influence of the sandbox environment, the reward modeling system, and the RL algorithm itself. For the RL method, we additionally log the sample keep rate and the loss-mask ratio throughout training. These metrics jointly inform training stability and failure modes: low entropy suggests poor exploration, while excessively high gradient norms indicate instability; unusually short responses and few turns often signal reward hacking; low tool-call accuracy and verifier success point to systematic execution or evaluation errors; and a very low sample keep rate typically means the data regime is misaligned (too easy or too hard), reducing the need—or opportunity—for exploration.

We show a training metric monitoring example in Figure 6. As can be seen, for agentic RL, the external environment is not always table (e.g., sometimes the verifier also will failed due to large-scale reward services, and the tool call accuracy is also not stable even we derive a offline sandbox). In addition, different traditional RL training, we didn't observe the monotonically decreasing entropy. This is because in agenti RL process, agent should deal with continually change tool response obtained from the environment. In addition, the interaction turn is also change over the environment.

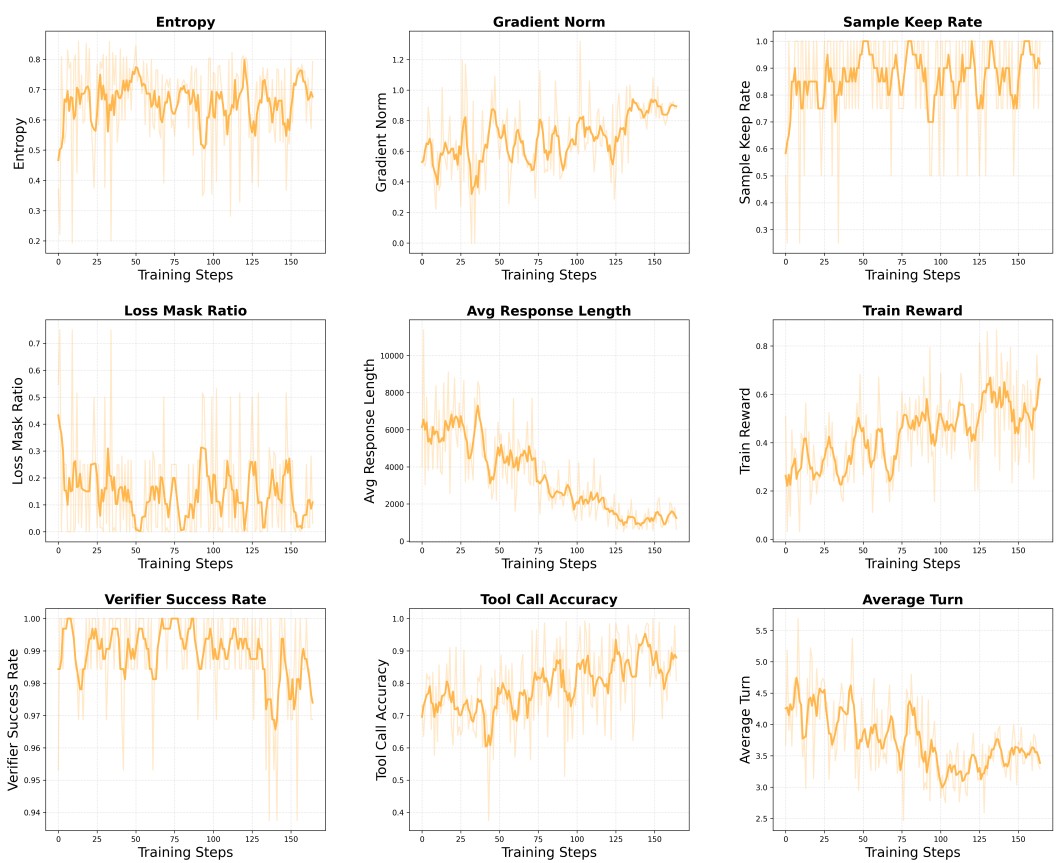

Figure 6: Metrics monitoring in RL training process. During agentic RL training, we periodically monitor a set of key indicators: policy entropy, training gradient norm, sample keep rate, loss-mask ratio, average response length, training reward, verifier success rate, tool-call accuracy, and average number of turns. Anomalies or regressions in any of these metrics can precipitate training failure, which also indicates the challenges of agentic RL.

## A.5 HUMAN ANNOTATION

As shown in Table 5, the human annotation process consists of seven dimensions designed to comprehensively assess the quality of AI-powered travel itinerary:

- **User Intention Understanding** evaluates the system's ability to correctly parse and interpret user inputs, ensuring all critical travel parameters are accurately captured. This dimension is fundamental as misunderstanding user requirements leads to irrelevant recommendations.

- **Itinerary Completeness** assesses whether the recommendation covers all essential travel components (flights, accommodation, local transportation) and maintains temporal coherence. A complete itinerary should provide seamless transitions between different travel segments.

- **Itinerary Feasibility** examines the practical executability of the proposed itinerary. This includes verifying that the schedule is not overly ambitious, transportation connections are realistic, and the overall route forms a logical closed loop.

- **Itinerary Affordability** focuses on the economic and practical aspects of the recommendation, ensuring resources are actually bookable, prices are accurate, and recommendations comply with organizational travel policies while maintaining cost-effectiveness.

- **Itinerary Clarity** measures both the efficiency of the reasoning process and the transparency of the reasoning provided. Quick responses with clear justifications to convince user.

- **Personalized Requirement** evaluates the system's capability to incorporate individual user preferences and historical patterns, ensuring recommendations align with user habits and preferences.

- **Without Factual Hallucination** serves as a critical safety check, identifying instances where the AI system generates non-existent services or logically inconsistent outputs that could mislead users.

Table 5: Evaluation dimensions for TP agent on human annotation.

| Evaluation Dimension | Explanation | Score | Scoring Criteria |
|---|---|---|---|
| User Intention Understanding | Whether accurately identifies user requirements: departure/destination, time, budget, travel scenario | TRUE/FALSE | **TRUE**: All elements correctly identified

**FALSE**: Missing or incorrect identification of key elements |
| Itinerary Completeness | Whether recommendation covers all key components: flights + hotels + transportation, with coherent timing | TRUE/FALSE | **TRUE**: Complete coverage, coherent

**FALSE**: Location/time deviations, incomplete components |
| Itinerary Feasibility | Whether the itinerary is practically executable with reasonable conditions | TRUE/FALSE | **TRUE**: Closed-loop, reasonable arrangement
**FALSE**: Overly tight schedule, unreasonable combinations, non-closed loop |
| Itinerary Affordability | Whether recommended resources are bookable, accurate pricing/inventory, compliant with company policy, cost-effective | TRUE/FALSE | **TRUE**: Real resources, reasonable pricing, policy compliant

**FALSE**: Slight exceedance (¿15%), unbookable options |
| Itinerary Clarity | Whether recommendation is fast (within 200s) with clear justification (e.g., "cost-effective", "close to meeting venue") | TRUE/FALSE | **TRUE**: Fast response with clear reasoning

**FALSE**: Slow response or vague/unexplainable justifications |
| Personalized Requirement | Whether recommendation considers user history/preferences (airline preference, time preference, geographic preference, accommodation type, etc.) | TRUE/FALSE | **TRUE**: Accurately matches user preferences

**FALSE**: Misses personalized tags or violates user habits |
| Without Factual Hallucination | Whether recommendation contains AI-fabricated content, such as non-existent flights/hotels, logical errors, etc. | TRUE/FALSE | **TRUE**: Factually correct recommendations

**FALSE**: Contains hallucinations or logical errors |

## A.6 CASE STUDY

Due to ethic requirement, we cannot release the data used for training. To ease reader understanding, we provide a real-world user query, and corresponding agent's response here.

Specifically, we select several real-world user case from the deployed travel planning service on DiDi Enterprise Solution App. The illustrative case hopes to facilitate the reader's understanding of how DeepTravel enable TP agent autonomously plan, execute tool, and reflect on tool response for exploring and refine its intermediate steps throughout multi-turn reasoning process.

As illustrated in Table A.6, the TP agent—powered by DeepTravel-32B—follows a multi-turn tool-integrated reasoning. Concretely: (1) Intent understanding and constraints grounding. (2) Tool planning: Before calling tools, the agent drafts a tool plan with time buffers and fallback branches. (3) Tool execution with schema-aligned calls: All tool calls are structured with explicit parameters and expected fields, ensuring reproducibility and traceability. (4) Tool-response interpretation and evidence binding: The agent parses returned fields and binds them to the generated itinerary via card-style evidence, avoiding hallucinations. (5) Reflection: If constraints are unmet (e.g., arrival risk due to transfer), the agent revises either the departure time window or the station choice, and re-calls tools with updated parameters. (6) Finalize the travel Itinerary: The agent assembles a coherent plan with justification, alternative options, and risk notes, each item linked to tool evidence.

1080
1081

### A.6 Reasoning Process of DeepTravel-32B

1082
1083

**User Query**: Travel from Suzhou to the Wuhan Conference Center on June 26th and need to arrive before 3:00 PM. I'd like to stay near the riverside area. Please design a travel plan..

1084
1085

**Agentic Travel Planning**:

1086
1087

```
<think>Ok. So, I first need to confirm the user's needs. The
user plans to travel from Suzhou to the Wuhan Conference Center
```

1088
1089

```
...
We also generate the corresponding card information to ensure
that the data comes from the tool's return results.
```

1090

```
</think>
```

1091
1092

```
<tool_call_thinking>Search for direct high-speed trains from
Suzhou to Wuhan, departing before 12:00 PM (assuming a 3-hour
```

1093
1094

```
...
and allowing 1 hour for city transportation)
```

1095

```
</tool_call_thinking>
```

1096
1097

```
<tool_call>train_search(depart_station="Suzhou", arrive_station
="Wuhan", depart_date="2025-06-26", is_transfer=0)
```

1098

```
</tool_call>
```

1099

```
<tool_response>...</tool_response>
```

1100

```
<tool_response_thinking>Several high-speed trains were found, but
...
```

1101
1102

```
</tool_response_thinking>
```

1102
1103

```
<tool_call_thinking>Search for hotels near Wuhan River Beach,
check-in date is June 26, check-out date is to be June 27.
```

1104

```
</tool_call_thinking>
```

1105
1106

```
<tool_call>hotel_search(city_name="Wuhan", checkin_date="2025-
06-26", checkout_date="2025-06-27")
```

1107

```
</tool_call>
```

1108

```
<tool_response>...</tool_response>
```

1109

```
<tool_response_thinking>I found several hotels near the river
...
```

1110
1111

```
</tool_response_thinking>
```

1111

```
<answer>
```

1112

```
### Outbound Recommendations (June 26, 2025)
```

1113

```
#### G3124 High-Speed Train (Suzhou → Wuhan)
```

1114

```
- **Recommendation**: Departs at 06:54, arrives at Wuhan Station
```

1115
1116

```
...
```

1117

```
#### D3056 High-Speed Train (Suzhou → Hankou)
```

1118

```
- **Recommendation**: Departs at 08:06, arrives at Hankou Station
```

1119

```
...
```

1120
1121

```
### Hotel Recommendations
```

1121

```
#### Qingchen Yuejiang Hotel (Jianghan Road Pedestrian Street)
```

1122
1123

```
- **Recommendation**: Only 233 meters from the river beach,
```

1123

```
...
```

1124
1125

```
#### Le Bo 1946 Riverside Hotel (Hankou Jiangtan, Lihuangpih)
```

1126

```
- **Recommendation**: River view rooms available, smart features
```

1127

```
...
```

1128
1129

```
**Friendly Tips**:
```

1130

```
1. Please book your shuttle in advance through the DiDi ES...
```

1131

```
2. Please confirm room availability and breakfast policies...
```

```
</answer>
```

1132
1133

