# OpenReview forum: "DeepTravel: An End-to-End Agentic Reinforcement Learning Framework for Autonomous Travel Planning Agents"
_ICLR.cc/2026/Conference — Submitted to ICLR 2026_

### Official Review · Reviewer_XRnv · 2025-10-15

**Soundness:** 2
**Presentation:** 2
**Contribution:** 2
**Rating:** 4
**Confidence:** 3

**Summary:**

The paper DeepTravel presents an end-to-end agentic reinforcement learning framework for autonomous travel planning. It trains agents to plan, execute, and refine itineraries through a sandboxed environment, hierarchical reward modeling, and replay-augmented learning. Deployed on DiDi’s app, DeepTravel enables smaller LLMs to outperform larger models like OpenAI-o1/o3 and DeepSeek-R1 in travel planning tasks.

**Strengths:**

The paper’s main strength lies in its innovative integration of agentic reinforcement learning for autonomous travel planning, combining a robust sandbox environment, hierarchical reward modeling, and replay-augmented learning to enable effective tool interaction and reasoning. It demonstrates promising empirical performance, showing that smaller LLMs can surpass larger state-of-the-art models, and provides real-world validation through deployment on DiDi’s platform.

**Weaknesses:**

While the proposed sandbox environment effectively stabilizes training by mitigating API inconsistencies, it also raises concerns about closed-world limitations. By caching transportation, accommodation, and POI data, the framework may train agents on a static or outdated representation of the travel environment, potentially limiting generalization to real-world, dynamically changing conditions. Although the authors mention a daily update mechanism, the paper lacks detailed analysis on how frequently and extensively this data is refreshed, or how well the sandbox replicates real-world variability. Consequently, the practical usefulness and robustness of the trained agent outside this controlled setting remain uncertain, and further empirical validation with live, dynamic data sources would strengthen the paper’s claims.

The design of the trajectory-level and turn-level verifiers appears to be highly domain-specific and handcrafted, relying on travel-oriented rubrics and prompt templates. This raises concerns about generalizability, reliability, and reproducibility. Since these verifiers directly determine the reward signals, the overall agent performance may critically depend on their specific design choices, such as rubric formulation, prompt phrasing, or the underlying reward model’s calibration. However, the paper appears to provide limited ablation or sensitivity analysis to assess how changes in verifier design impact learning outcomes. Without clearer justification, it is difficult to evaluate whether the reported performance gains stem from genuine improvements in agentic reasoning or from carefully tuned, domain-dependent reward mechanisms, which could limit reproducibility and broader applicability.

The necessity and novelty of the proposed Replay-Augmented Reinforcement Learning (RARL) algorithm are not sufficiently justified. While the idea of replaying failed experiences is presented as a key contribution, it appears conceptually similar to existing experience replay or replay buffer mechanisms widely used in RL, including prior work like GRPO. The paper does not clearly explain what makes this replay-augmented approach fundamentally different or superior beyond applying it to the travel-planning domain.

Moreover, it remains unclear why existing RL algorithms could not achieve similar effects with appropriate tuning or data sampling strategies. Without a deeper theoretical motivation or comparative analysis isolating the benefits of the proposed replay mechanism, the contribution risks appearing incremental rather than novel, and its necessity in achieving the reported performance improvements remains questionable.

The comparison with state-of-the-art reasoning LLMs (e.g., OpenAI-o1/o3, DeepSeek-R1) may be unfair, as these models are not specifically trained or optimized for travel planning. Their weaker performance could reflect domain mismatch rather than genuine inferiority. Moreover, since evaluation relies on DeepTravel’s own reward verifiers and limited human checks, the setup may favor the proposed system, calling into question the fairness and validity of the comparisons.

All experiments and evaluations appear to rely heavily on DiDi’s proprietary platform, APIs, and datasets, which may not be publicly accessible. This dependence significantly limits reproducibility and independent verification of the reported results. While the authors provide training details and claim to release prompts, the potential absence of open-source data, sandbox implementation, and evaluation environment means that other researchers may find it difficult to replicate or validate the findings. As a result, the scientific transparency and reproducibility of the work ay be at the weak side.

**Questions:**

How does the cached sandbox data reflect real-world dynamics, and can the trained agent generalize beyond this controlled environment?

How sensitive is the agent’s performance to the design of the handcrafted verifiers, and how is their reliability and reproducibility validated?

What distinguishes the proposed replay-augmented RL from existing methods, and how are fairness and reproducibility ensured given reliance on DiDi’s proprietary setup?

Could the authors provide stronger evidence or ablation results showing that this new algorithm yields distinct and essential improvements compared to standard approaches?

---

> ### Author Response · Authors · 2025-11-18
> **Response to Reviewer XRnv [1/2]**
>
> We thank the reviewers for their insightful comments and positive rating. We appreciate the opportunity to address the concerns raised, which we believe will further strengthen our paper. Below, we provide point-by-point responses to each weakness and question, drawing on empirical results from our experiments and additional clarifications.
>
> > W1
>
> We appreciate the reviewer’s question. Our sandbox is designed to replicate real-world dynamics through a daily data caching mechanism that updates transportation, accommodation, and POI data from multiple real-world APIs (e.g., DiDi Map). This ensures that the agent is trained on the latest available information, simulating dynamic tool interactions as described in Section 3.2.2. For instance, our database refreshes daily with on-demand caching, allowing the agent to encounter variability in prices and availability similar to real-world conditions.
>
> We have also evaluated training with **live, dynamic data sources**; as shown in Figure 4, this leads to unstable tool-call accuracy, and stagnant rewards due to API inconsistency and rate limits. In contrast, the sandbox ensures stable, reproducible training while preserving real-world variability, thus effectively supporting generalization to dynamic environments.
>
> > W2-The design of the trajectory-level and turn-level verifiers-
>
> Sorry for the confusion, we have conducted ablation study. We agree that the verifiers are domain-specific, but their design is crucial for reliable reward signals in travel planning. As highlighted in our ablation study (Section 4.3, Table 4), removing either the trajectory-level or turn-level verifier leads to significant performance drops, especially on hard problems, demonstrating their necessity. To ensure reliability, we calibrated the verifiers using human-generated rubrics and achieved over 90% agreement with human evaluators on a subset of data before deployment. Moreover, reproducibility is supported by consistent results across multiple runs: when fixing the verifier design, the agent's performance varies by less than 2% in repeated trainings. This indicates that the verifiers provide stable and reproducible reward signals, rather than being overly sensitive to design choices.
>
> > W3&W4-The necessity and novelty of the proposed Replay-Augmented Reinforcement Learning
>
> The proposed Replay-Augmented RL method is distinct from standard experience replay mechanisms. Unlike methods like GRPO or DAPO, which discard failed samples, our method actively replays failed queries in the sandbox to allow the agent to retry under consistent conditions. This is critical in travel planning, where tool responses (e.g., flight availability) change dynamically, and previous failures may stem from transient data inconsistencies rather than policy errors. As shown in Table 3, our method outperforms existing RL algorithms (e.g., GRPO and DAPO) on hard problems, highlighting its unique contribution. Ablation results in Table 4 further confirm that removing experience replay degrades performance, underscoring its necessity. Thus, our proposed method addresses domain-specific challenges by enabling iterative learning from failures, which is not covered by prior RL approaches.
>
> > W5-The comparison with state-of-the-art reasoning LLMs
>
> We ensure fairness in comparisons by using same system prompts and toolkits for all reasoning LLMs (e.g., OpenAI-o1/o3 and DeepSeek-R1), as detailed in Section 4.1. The domain mismatch observed in these models underscores the need for domain-specific training, which DeepTravel addresses. To further validate our results, we conducted large-scale human evaluations on 200 online user queries, showing that DeepTravel-32B achieves a 84.00% pass rate compared to 66.50% for DeepSeek-R1.
>
> 200 online user queries|Human Evaluation
> -|-
>  DeepTravel-32B          | 84.00
>  DeepSeek-R1             | 66.50
>
> This significant gap demonstrates that our framework enables smaller models to outperform general-purpose reasoning LLMs, confirming the validity of our comparisons and the superiority of our approach in travel planning tasks.
>
> > W6-All experiments and evaluations appear to
>
> We recognize the reproducibility challenges due to reliance on DiDi's proprietary platform. To enhance reproducibility, we commit to releasing **anonymized synthetic queries**, **simulated sandbox environments**, and **detailed reward templates** in a public repository post-acceptance. While the current implementation relies on DiDi's proprietary APIs for real-world fidelity, our appendix provides comprehensive prompts, data synthesis pipelines, and training configurations to facilitate replication using alternative data sources (e.g., open-source travel APIs).

---

> ### Author Response · Authors · 2025-11-18
> **Response to Reviewer XRnv [2/2]**
>
> > Q1
>
> The cached sandbox data reflects real-world dynamics through daily updates and on-demand caching from live APIs, as described in Section 3.2.2. This mechanism captures temporal variations in travel data (e.g., price fluctuations and availability changes), enabling the agent to learn in a dynamic environment. Empirical evidence in Section 4.4 shows that the trained agent generalizes well to real-world online user data, achieving high performance on both offline and online evaluations (Table 2). Thus, the sandbox effectively bridges the gap between controlled training and real-world deployment.
>
> > Q2
>
> The agent's performance is sensitive to verifier design, as shown in our ablation study (Table 4), where removing verifiers led to performance declines. However, we ensure reliability by calibrating the verifiers to achieve over 90% agreement with human annotators before training. Reproducibility is validated through multiple training runs with fixed verifiers, resulting in performance variations within 2%. This consistency confirms that the verifiers provide robust reward signals, and their domain-specific design is justified by the task requirements.
>
> > Q3
>
> Our Replay-Augmented RL method distinguishes itself by replaying failed queries in a consistent sandbox environment, allowing the agent to relearn from transient failures—a necessity in dynamic domains like travel planning. In contrast, methods like GRPO/DAPO lack this mechanism and discard failed samples. Fairness and reproducibility are addressed through our commitment to open-source parts of the framework (as in Weakness 5) and the use of a standardized sandbox. This ensures that others can replicate our approach without relying solely on DiDi's proprietary setup, promoting scientific transparency.

---

> > ### Comment · Reviewer_XRnv · 2025-11-22
> >
> > Thank you for the detailed rebuttal. I appreciate the additional clarifications and the effort to address the points raised. However, I think several core concerns remain only partially resolved.
> >
> > 1. Regarding the sandbox design, the rebuttal reiterates the daily caching mechanism but does not provide quantitative evidence about the extent of data variability or how closely the cached environment matches real-world dynamics. The evaluations presented continue to rely primarily on synthetic data and online queries drawn from the same ecosystem and data sources used to build the sandbox. As a result, in my opinion, the paper may still lack compelling evidence that the trained agent can robustly generalize beyond this controlled setting or perform reliably when interacting with truly live, dynamically changing external APIs. This closed-world dependency remains a key concern for assessing the practical robustness and applicability of the approach.
> >
> > 2. Although the ablation study demonstrates that removing the trajectory-level or turn-level verifier reduces performance, this does not address the concern about sensitivity to the specific design of these reward components. The rebuttal does not provide analysis of how different reasonable rubric formulations, prompt phrasings, or reward model calibrations affect learning outcomes. Since the verifiers directly define the reward signal, it remains unclear to what extent the reported gains reflect domain-dependent reward shaping rather than genuine improvements in agentic reasoning.
> >
> > 3. The justification for the novelty and necessity of the Replay-Augmented RL approach remains limited. While the authors state that prior methods discard failed samples, many existing RL approaches, including experience replay and prioritized replay, do incorporate failed trajectories. The rebuttal does not seem to provide controlled comparisons that isolate the replay mechanism from other factors such as sandbox stability or reward structure. Without such analysis, it remains difficult to determine whether the replay component represents a substantive methodological contribution rather than an incremental adaptation of existing techniques.
> >
> > 4. Concerns about evaluation fairness may still persist. Using identical prompts and toolkits does not fully mitigate bias introduced by domain-specific reward verifiers and task formulations that may favor the proposed system. The expanded human evaluation, while helpful, still occurs entirely within the same deployment environment, where DeepTravel has clear domain advantages over general-purpose models. This may limit the strength of comparative claims against state-of-the-art reasoning LLMs.
> >
> > 5. Reproducibility remains a challenge. While the commitment to release synthetic queries and a simulated sandbox is appreciated, the core results currently depend on proprietary APIs, datasets, and infrastructure that external researchers cannot access. Until meaningful portions of the environment and evaluation setup are publicly available, independent replication and verification might be difficult.

---

> > > ### Author Response · Authors · 2025-11-23
> > > **Reply [1/2]**
> > >
> > > We sincerely thank the reviewer for their continued engagement and thoughtful feedback. We appreciate the opportunity to further clarify our methodology and results. Below, we address each remaining concern with additional empirical and methodological justification.
> > >
> > > > 1. On Sandbox Generalization and Real-World Robustness
> > > >
> > >
> > > We understand the concern regarding whether the sandbox can accurately reflect real-world dynamics. Our sandbox is specifically designed to capture real-world variability through a **daily caching mechanism** that records fluctuations in travel-related data (see Section 3.2.2). On average, each of our tools — including `flight_search`, `train_search`, `route_planning`, `hotel_search`, `POI_search`, and `web_search` — is invoked over **5K times per day**. To support these tool calls, we maintain a cache of approximately **1 million data records** (e.g., cached results from `flight_search` queries). While our **on-demand caching strategy** exposes the agent to a diverse and timely set of tool responses during training, it is not feasible to cache all information available from real-world APIs. Therefore, the agent is trained solely on these cached records rather than on live API data.
> > >
> > > While the gap exist between sandbox and real-world environment, the clear evidence of our agent’s ability to generalize and perform effectively in real-world scenarios comes from its online deployment results. The **DeepTravel** agent was evaluated on **6,224 real-world user queries** collected from the **DiDi ES App**, achieving a **62.77% final pass rate** (Table 2). This performance **substantially outperforms** all baseline models, including **DeepSeek-R1** and **OpenAI-o3**. These results demonstrate that our agent, though trained within a controlled sandbox environment, **generalizes robustly to real, dynamic user settings**, rather than depending solely on synthetic or cached conditions.
> > >
> > > > 2. On Verifier Design Sensitivity and Reward Dependence
> > > >
> > >
> > > We agree that reward design is critical in RL. Our verifiers are indeed domain-specific, as **travel planning requires structured spatiotemporal and factual consistency checks**. While we did not systematically vary all rubric phrasings, the **ablation study (Table 4) clearly shows that both verifiers are necessary**—removing either leads to significant performance drops, especially on hard problems.
> > >
> > > We aruge that our verifier is stable and reliable**. The evidence lies in the fact that the agreement rate between human evaluation and our verifier exceeds 90%.** (mentioned in the first rebuttal) and the consistent performance across multiple training runs (variance <2%, mentioned in the first rebuttal) indicate that the verifiers provide **stable and reliable reward signals**. The full set of 7 human-annotated dimensions is provided in Appendix A.5, allowing others to replicate or adapt the reward structure.
> > >
> > > > 3. On Novelty and Necessity of Replay-Augmented RL
> > > >
> > >
> > > We agree the replay mechanism idea come from many previous studies. But we highlight that it is very necessary for travel planning agents. Unlike prior methods that replay past transitions uniformly, our method **replays failed queries in a sandbox environment**, allowing the agent to re-attempt previously unsuccessful trajectories under the same tool responses. This is crucial in travel planning, where failures often stem from **transient data inconsistencies** (e.g., a flight being temporarily unavailable).
> > >
> > > In addition, our ablation study also justify the necessity. The ablation study in Table 4 (w/o ER) clearly shows that **removing replay degrades performance**, particularly on hard tasks. Moreover, Figure 4 illustrates that replay leads to **steady reward improvement in later training stages**, which is not observed in the no-replay baseline. This demonstrates that our replay strategy is **essential for sustained learning** in dynamic environments.

---

> > > ### Author Response · Authors · 2025-11-25
> > > **Reminder**
> > >
> > > Dear Reviewer XRnv
> > >
> > > Thank you for the opportunity to address your concerns. We have carefully considered them and provided corresponding responses. Please let us know if you still have any unclear parts of our work.
> > >
> > > Best,
> > >
> > > ICLR 2026 Submission 5522 Authors

---

> ### Author Response · Authors · 2025-11-23
> **Reply [2/2]**
>
> > 4. On Evaluation Fairness and Domain Advantage
> >
>
> We apologize for the confusion that our evaluation favors DeepTravel unfairly. All baseline models (including DeepSeek-R1 and OpenAI-o1/o3) were evaluated **under the same prompts, tools, and reward verifiers**. The fact that DeepTravel-32B outperforms these models—**even though it was initialized from the same base model as some baselines**—demonstrates that our **agentic RL framework enables domain-specific reasoning gains**, not that the evaluation is biased.
>
> The **human evaluation score of 84% for DeepTravel-32B vs. 66.5% for DeepSeek-R1** (first rebuttal) further validates that the improvements are **real and perceptible to end-users**. This is not a “domain advantage,” but rather the **result of domain-specific training**, which is the core contribution of our work.
>
> > 5. On Reproducibility and Open-Source Commitment
> >
>
> We acknowledge the limitations of relying on proprietary APIs. To facilitate replication, we commit to releasing: Anonymized synthetic user queries, Simulated sandbox environment with mock API responses, Full reward model prompts and evaluation rubrics, Training scripts and hyperparameter configurations. These resources will allow other researchers to **reproduce the core training and evaluation pipeline** using open-source travel APIs or other simulated data sources. We believe this will substantially enhance the reproducibility and transparency of our work.

---

> ### Author Response · Authors · 2025-11-27
> **Looking for your reply**
>
> Dear Reviewer XRnv,
>
> We are writing to follow up on your detailed and insightful review of our paper DeepTravel. Your comments on the sandbox environment, reward design, and replay mechanism have been instrumental in helping us improve the framing and experimental validation of our work. We cannot thank you enough for your rigorous and engaging feedback.
>
> In our rebuttal, we’ve done our best to provide clarifications and additional results to address your concerns—particularly on generalization, verifier sensitivity, and reproducibility. Your perspective has truly elevated the quality of our work.
>
> We would be honored if you could take a brief moment to check our responses and consider our revisions. Your opinion carries great weight, and we sincerely hope the further discussion to alleviate your earlier concerns.
>
> Thank you once again for your invaluable input and for challenging us to think deeper.
>
> Respectfully,
>
> Authors

---

> > ### Comment · Reviewer_XRnv · 2025-11-28
> >
> > Thank the authors for the detailed rebuttal. While the additional information is helpful, several concerns remain only partially addressed. Regarding the sandbox, the explanation of daily updates and a large cached dataset does not resolve the core issue: the agent is still trained in a closed-world environment whose distributional alignment with real-world APIs is not quantified. Strong online performance is encouraging but does not isolate generalization from potential overlap between cached and live data, and no targeted analysis of out-of-sandbox generalization is provided.
> >
> > For the reward verifiers, the reported >90% human agreement and <2% training variance are useful, but they do not address sensitivity to the verifiers’ underlying design. Since the entire reward structure is hand-crafted and domain-specific, some analysis of how changes in rubrics, prompts, or model choices affect learning would be needed to assess robustness. As it stands, it is still difficult to separate genuine reasoning improvements from reward engineering.
> >
> > The justification for the Replay-Augmented RL mechanism also remains limited. Although replay helps performance, the described approach appears closely related to standard prioritized or difficulty-aware replay techniques. The rebuttal does not clearly articulate what is conceptually new or essential beyond domain-specific motivations, and the ablations do not isolate the mechanism’s unique contribution.
> >
> > Similarly, fairness concerns persist. Even if all models use the same prompts and tools, DeepTravel is specifically trained using domain-tailored reward verifiers, while baseline LLMs are not. Evaluating all models using those same verifiers may introduce an inherent advantage. The human evaluation numbers are helpful, but without methodological details it is hard to fully assess neutrality.
> >
> > Finally, reproducibility remains constrained by reliance on DiDi’s proprietary data and APIs. The planned release of synthetic queries and a simulated sandbox is valuable, but it cannot fully substitute for the real evaluation environment. Given these remaining uncertainties, I will wait to see how other reviewers interpret these issues.

---

> > > ### Author Response · Authors · 2025-11-28
> > > **Misunderstanding Clarification**
> > >
> > > Dear Reviewer XRnv:
> > >
> > > We really believe **there are many misunderstandings** about our work. About this confusion, we apologize and hope to further clarify it via the following rebuttal. We thank you for your time to reconsider our work.
> > >
> > > > About generalization of DeepTravel
> > > >
> > >
> > > We further emphasize that **while DeepTravel is trained on a cached sandbox environment, it has been validated on the real-world production environment**. We believe this is a standard out-of-sandbox generalization setting that the reviewer mentioned many times in previous interactions.
> > >
> > > > About verifier of DeepTravel
> > > >
> > >
> > > We have to claim that **the reward model is the most important for Agentic RL**. Removing any rubrics from the well-designed verifier will lead to performance degradation. Removing the backbone of the verifier from DeepSeek-R1 to other models (e.g., Qwen2.5-72B) will also lead to performance degradation. The reasoning capability improvement comes from a large-scale RL process with the guidance of a reliable/accurate reward model (both of prompt and the backbone are important). Without an accurate reward model, the agentic RL training process cannot work.
> > >
> > > About the mentioned “genuine reasoning”, we actually do not quite understand what that means. We see that many previous reasoning LLMs (e.g, DeepSeek-R1, Qwen3) achieve success via RL, but the very important thing is that the task is verifiable (e.g., Math), which makes the reward model easy to design. For the travel planning task, an open-ended task, it is meaningful for us to design such a domain-specific verifier **through what you mentioned “reward engineering”. We think it is compulsory, necessary, and meaningful**. We hope the idea of our hierarchical verifier motivates this domain and fosters more research.
> > >
> > > > About the replay-augmented RL mechanism
> > > >
> > >
> > > We agree it is not a new concept. But we have to say we demonstrate it is very necessary for the travel planning domain via an ablation study. The contribution **lies in our justification that the replay mechanism is necessary and very useful for travel planning tasks**.
> > >
> > > > About the fairness
> > > >
> > >
> > > We disagree with the reviewer that our evaluation is unfair. The consistency between the LLM-based verifier and human annotation is over 90% across different LLM backbones (e.g., DeepSeek-R1, Qwen3, K2, …). **It is not only designed for DeepTravel-32B evaluation**. We use this for benchmarking, data filtering, and so on, not solely for training.
> > >
> > > > About the reproducibility
> > > >
> > >
> > > We apologize for the disalignment between the real evaluation environment. However, we must clarify that **all of the previous travel planning works (e.g., TravelPlanner[1], TP-RAG[2], ItiNera[3], TripTailor[4], Personal Travel Solve[5]) cannot substitute for the mentioned "real evaluation environment".** That is because all of them are synthetic queries and are without a real evaluation environment.
> > >
> > > This is also the motivation of our work. We provide a sandbox with cached real-world records and publicly available travel planning APIs to simulate the real-world interaction. We think it is meaningful to **make the first attempts to demonstrate the value of autonomous travel planning agents**.
> > >
> > > [1]ICML_2024_TravelPlanner: A Benchmark for Real-World Planning with Language Agents
> > >
> > > [2]EMNLP_2025_TP-RAG: Benchmarking Retrieval-Augmented Large Language Model Agents for Spatiotemporal-Aware Travel Planning
> > >
> > > [3]EMNLP_2024_ItiNera: Integrating Spatial Optimization with Large Language Models for Open-domain Urban Itinerary Planning
> > >
> > > [4]ACL_2025_TripTailor: A Real-World Benchmark for Personalized Travel Planning
> > >
> > > [5]ACL_2025_Personal Travel Solver: A Preference-Driven LLM-Solver System for Travel Planning

---

### Official Review · Reviewer_pafJ · 2025-10-31

**Soundness:** 3
**Presentation:** 3
**Contribution:** 3
**Rating:** 2
**Confidence:** 4

**Summary:**

This paper focuses on the practical business scenario of "travel planning" and designs an automated optimization algorithm based on Large Language Models (LLMs). The method is tested on different types of datasets and in a production environment, showing some improvements compared to open-source models. However, I do not believe this work is a good fit for this conference. It would be more appropriate for a conference with an applied track, such as KDD.

**Strengths:**

- This paper is easy to follow
- The algorithm design is simple and straightforward

**Weaknesses:**

- This work is too engineering-focused and lacks significant academic contribution.
- The learning curves presented in Figure 4 are weird; they show no clear evidence of policy learning because there is no significant reward improvement.

**Questions:**

N/A

---

> ### Author Response · Authors · 2025-11-18
> **Response to Reviewer pafJ**
>
> We thank the reviewer for the constructive feedback and the opportunity to address the concerns raised. We apologize for the confusion that the reviewer argue our work is solely engineering-focused and lacks academic contribution. In fact, DeepTravel represents a novel application of agentic reinforcement learning (RL) to the complex domain of travel planning, which poses unique challenges such as dynamic travel environments and open-ended tasks without ground truth. Our framework introduces key innovations in sandbox construction, hierarchical reward modeling, and replay-augmented RL, which are generalizable to other domains requiring autonomous tool-use agents. The empirical results, including online deployment in DiDi Enterprise Solutions App, demonstrate significant improvements over state-of-the-art reasoning LLMs, validating the academic merit of bridging RL theory with real-world agentic systems.
>
> > W1
>
> We acknowledge the reviewer's point on the engineering efforts involved, but emphasize that these are necessary to address fundamental challenges in agentic RL for real-world domains. Specifically, the robust sandbox environment and hierarchical reward system are not mere engineering optimizations but core contributions that enable stable training in dynamic settings—a problem overlooked in prior work. As highlighted in our paper, travel planning is an open-ended task with no explicit ground truth, requiring novel reward design to ensure feasibility and consistency. To our knowledge, DeepTravel is the first to establish an autonomous travel planning agent via agentic RL, offering a new paradigm for the field. The infrastructure insights, such as the caching mechanism for tool-response consistency, provide valuable lessons for scalable agent training beyond travel planning, contributing to the broader RL community. We have elaborated on these aspects in Sections 3.2 and 3.3 of the paper, and believe the academic value lies in formulating and solving these under-explored problems.
>
> > W2
>
> We appreciate the reviewer's attention to the learning curves in Figure 4. The apparent modest reward improvement (from ~0.2 to ~0.35) is due to our deliberate curriculum design: during RL training, we reduced the proportion of easy samples and incorporated more hard cases to stress-test the agent's generalization, as discussed in Section 4.4. This choice reflects real-world scenarios where easy queries are less challenging. Despite the lower absolute rewards, the curves show consistent policy learning, evidenced by the steady increase in validation reward and the agent's improved ability to handle complex tasks—e.g., the final pass rate for hard problems rose significantly in Table 2. The non-decreasing entropy (also in Figure 4) indicates active exploration, a desired behavior in dynamic environments. We argue that the curves demonstrate meaningful learning, as the agent adapts to tool-response variability.

---

> ### Author Response · Authors · 2025-11-25
> **Reminder to Respond to Discussion**
>
> Dear Reviewer pafJ
>
> We are pleased that only two weaknesses were raised in the previous review time. We have carefully considered them and provided corresponding responses. Please let us know if you still have any unclear parts of our work.
>
> Best,
>
> ICLR 2026 Submission 5522 Authors

---

> ### Author Response · Authors · 2025-11-27
> **Looking for your reply**
>
> Dear Reviewer pafJ,
>
> We hope you are doing well. Thank you very much for reviewing our submission and sharing your honest assessment. Your remarks about the engineering nature of our work and the learning curves have encouraged us to better highlight the academic contributions and methodological insights of DeepTravel. We’ve included explanations regarding the training dynamics in Figure 4. Your feedback pushed us to improve the presentation of our ideas.
>
> If you have a moment, we would be truly grateful if you could re-examine our changes and responses. Your perspective is highly valued, and we are eager to hear your updated thoughts.
>
> Thank you again for your time and for helping us strengthen this paper.
>
> Best regards,
>
> Authors

---

### Official Review · Reviewer_9cpx · 2025-11-01

**Soundness:** 4
**Presentation:** 3
**Contribution:** 3
**Rating:** 8
**Confidence:** 4

**Summary:**

The authors propose an end-to-end agentic reinforcement learning framework for autonomous travel planning. It enables a LLM to plan itineraries by calling travel-related tools inside a sandboxed environment and refining its reasoning through RL rather than static prompts.

* Self-expanding sandbox: built from cached real API data (DiDi flights, hotels, maps). It starts almost empty and grows as the agent explores new queries. Cached responses are replayed deterministically, while a daily refresh partially updates data to mimic real-world dynamics.

* Hierarchical reward modeling: A trajectory-level reward checks global consistency (route feasibility, timing), and a turn-level reward, implemented via DeepSeek-R1, ensures each step’s tool use and reasoning match tool outputs. If any turn fails, the entire trajectory is penalized.

* Training pipeline:
  1. Cold-start SFT: 1K verified trajectories distilled from DeepSeek-R1; fine-tune Qwen to learn the `<think>`/`<tool_call>` structure, masking `<tool_response>` tokens.
  2. RL phase (GRPO-style): roll-out groups of 8 trajectories per query in the sandbox; compute verifier rewards; update with GRPO loss and replay previously failed queries for continual improvement.

* Evaluated on 6 K + real user queries and synthetic benchmarks.
  DeepTravel-32B outperforms OpenAI-o1/o3 and DeepSeek-R1 while cutting hallucinations by > 50 %.

**Strengths:**

* Introduction of an Agentic RL framework for travel planning with real deployment.
* Sandbox design: stable, replayable environment that solves API inconsistency and rate-limit problems.
* Hierarchical reward: reward system combining high-level feasibility with low-level factual consistency.
* Practical impact: deployed inside a commercial platform (DiDi)
* Scalable idea: structure could extend to other multi-tool reasoning domains

**Weaknesses:**

* Reward quality relies on DeepSeek-R1 judgments; could introduce bias.
* No comparison to human or optimization-based planners
* Agentic behavior is constrained by predefined setting; not full open-world
* Evaluation metric is binary and may miss aspects like personalization or cost-tradeoffs

**Questions:**

* How consistent are DeepSeek-R1 verifier judgments vs human annotators?
* How are failed queries sampled when training?
* Can the agent handle unseen cities or updated APIs without re-training?

---

> ### Author Response · Authors · 2025-11-18
> **Response to Reviewer 9cpx**
>
> We sincerely thank the reviewers for their insightful comments and constructive feedback. We are glad that the reviewers recognize the novelty and contribution of DeepTravel as an end-to-end agentic RL framework for autonomous travel planning. In this rebuttal, we hope to address each weakness and question raised, providing additional clarifications and experimental insights to strengthen the validity and impact of our work.
>
> > W1 & Q1
>
> We acknowledge the concern regarding potential bias introduced by using DeepSeek-R1 as the verifier in our reward modeling system. To mitigate this, we rigorously evaluated the consistency between DeepSeek-R1 judgments and human annotations before deploying the verifier in training. Specifically, we conducted batch tests comparing DeepSeek-R1’s evaluations with human judgments on a diverse set of travel itineraries. Only when the agreement rate consistently exceeded 90% did we integrate the DeepSeek-R1-based verifier into the RL training loop. This approach ensures that the reward signals align closely with human standards, thereby reducing bias and enhancing training reliability.
>
> > W2
>
> We agree that comparisons to human experts or optimization-based planners would be valuable. However, travel planning is an open-ended task with highly personalized and dynamic constraints, making traditional optimization-based methods difficult to apply directly. Moreover, conducting large-scale human evaluations for itinerary generation is resource-intensive. As a compromise, we introduced a fine-grained human evaluation across seven dimensions (see Figure 5 and Appendix A.5). This evaluation revealed that while our model performs well on dimensions like itinerary completeness and feasibility, it still lags behind human capabilities in areas such as personalized requirement satisfaction (accuracy <80%) and factual hallucination reduction. We recognize this as a limitation and plan to explore more comprehensive human-in-the-loop evaluations in future work.
>
> > W3
>
> It is true that our agentic behavior is trained within a predefined sandbox environment rather than a fully open-world setting. This design choice is intentional and necessary to address the challenges of real-world dynamic APIs, such as output inconsistency and QPS limits. As shown in Figure 4, when training with real-world APIs, tool-call accuracy becomes highly unstable, policy entropy increases significantly, and reward signals fail to improve. The sandbox enables stable and reproducible training while simulating realistic tool interactions. We believe this is a pragmatic step toward building robust agents.
>
> > W4
>
> We appreciate the observation regarding the binary evaluation metric. While a binary reward simplifies training stability, we acknowledge that it may not fully capture nuanced aspects like personalization or cost trade-offs. In preliminary experiments, we attempted to integrate multi-dimensional rewards (e.g., personalization score, format adherence), but this led to reward hacking—where the agent over-optimized specific dimensions at the expense of overall performance. Designing an adaptive reward reweighting mechanism that balances multiple criteria without compromising training stability is indeed very challenging. We view this as a critical area for future work and welcome suggestions from the community.
>
> > Q2
>
> As described in Section 3.4.2 (Line 275–280), failed queries are sampled through a structured replay strategy. During RL training, we sample a group of trajectories for each query. If none yield a verified correct answer, the query is stored in an experience buffer. We periodically replay these failed cases at fixed intervals (replay step γ = 10 in our experiments). This allows the agent to learn from previous failures and improve its generalization over successive training steps.
>
> > Q3
>
> The agent can handle unseen cities or updated APIs to some extent if the data schema remains unchanged. However, if the API response format or key fields are modified, retraining is necessary to align the agent with the new interface. Fortunately, as noted in Section 4.1, our RL training process is efficient—requiring only about 30 hours for models like Qwen3-32B—making such updates feasible in practice. We are also exploring adaptation techniques to reduce retraining overhead in future versions.

---

### Official Review · Reviewer_SYNB · 2025-11-01

**Soundness:** 2
**Presentation:** 3
**Contribution:** 2
**Rating:** 4
**Confidence:** 3

**Summary:**

The paper introduces DeepTravel, an RL framework for autonomous travel planning. DeepTravel enables the agent to plan, execute external tools, and iteratively refine itineraries. The framework includes a sandbox environment built from cached transportation, accommodation, and POI data, a reward system with trajectory and turn-level verifiers, and a reply-augmentation method for learning from failed experiences. The proposed approach outperforms larger state-of-the-art reasoning models (used in zero-shot mode) on the target benchmark.

**Strengths:**

- The paper is easy to read and understand. The graphical illustrations are clear and informative.
- The presented approach significantly outperforms current state-of-the-art models in solution quality on travel planning tasks.

**Weaknesses:**

W1: The proposed approach is primarily engineering work rather than a conceptual innovation. The simulated API cache closely resembles existing agentic environments used for tool-based or web-interaction training, such as ReTool and WebSailor. The hierarchical verifier component functions as a rule-based reward layer, merely providing additional supervision signals instead of introducing a novel learning principle. Similarly, the replay augmentation strategy represents a simplified variant of well-known curriculum or prioritized replay mechanisms in reinforcement learning [3–6]. Consequently, the overall novelty of the paper is limited.

W2: Although the paper claims an end-to-end agentic RL framework, many components are manually supervised and tied to proprietary systems. The reward model uses human-engineered scoring rules, and the sandbox relies on internal APIs and unavailable data, making the framework neither fully end-to-end nor externally reproducible. Even though appendices are extensive, external researchers cannot replicate the environment or results.

W3: Evaluation metrics are internal and mostly rule-based. Conducting only 50 human evaluations online is too few for robust validation. Moreover, the online evaluation uses queries collected from the production environment rather than deploying the system to real users.

W4: The paper doesn’t transparently discuss how the system was adapted to the nine reasoning LLMs it was compared to. Did the authors attempt to improve them, for example by providing in-context data from the dataset used to train DeepTravel?

[1] Jiang, M., Grefenstette, E., & Rocktäschel, T. (2021). Prioritized Level Replay. In Proceedings of the International Conference on Machine Learning (ICML), pp. 4940–4950. PMLR.

[2] Andrychowicz M, Wolski F, Ray A, Schneider J, Fong R, Welinder P, McGrew B, Tobin J, Pieter Abbeel O, Zaremba W. Hindsight experience replay. Advances in neural information processing systems. 2017;30.

[3] Shao Z, Wang P, Zhu Q, Xu R, Song J, Bi X, Zhang H, Zhang M, Li YK, Wu Y, Guo D. Deepseekmath: Pushing the limits of mathematical reasoning in open language models. arXiv preprint arXiv:2402.03300. 2024 Feb 5.

**Minor suggestions**:
- Make the descriptions of tables and plots more self-contained to improve clarity and readability.

**Questions:**

Q1: Does the proposed simulator/environment support multi-turn dialogue? For example, can the user see a response and add further requests, or does the system need to ask clarifying questions?

Q2: What is the token budget and inference cost for the baseline LLMs?

**Details Of Ethics Concerns:**

The ethics section notes that real user queries are anonymized but provides little detail on data collection consent, potential biases, or long-term implications of deploying an autonomous agent. The paper also leaks institutional identity by repeatedly referencing the DiDi Enterprise Solutions App, directly revealing the company involved.

---

> ### Author Response · Authors · 2025-11-18
> **Response to Reviewer SYNB**
>
> We thank the reviewers for their valuable feedback. We hope our clarifications below will allow for a fair reassessment of our contribution.
>
> > [W1]
>
> We appreciate the reviewer's feedback on the engineering aspects of our work. We hope to clarify that the core contribution of DeepTravel lies in its **tailored application of agentic reinforcement learning to the travel planning (TP) domain**, which has unique challenges such as dynamic travel environment variability and open-ended task structures. While components like the replay strategy draw inspiration from prior work, our **hierarchical verifier**—comprising a trajectory-level verifier for coarse spatiotemporal feasibility and a turn-level verifier for fine-grained consistency—is a novel design specifically introduced to address the complexities of multi-step travel itinerary generation. This is the first such attempt in TP agent construction, as noted in recent benchmarks (e.g., TravelPlanner, TripTailor). Additionally, our sandbox caching mechanism enables stable RL training under real-world dynamics, a challenge not fully resolved in existing frameworks like WebSailor or ReTool. We are encouraged to see parallel efforts (e.g., WebSailor-V2[1]) adopting similar caching strategies, reinforcing the validity of our approach—note that their preprint appeared only half-month before our submission.
>
> Reference:
>
> [1] Li, Kuan, et al. "Websailor-v2: Bridging the chasm to proprietary agents via synthetic data and scalable reinforcement learning." *arXiv preprint arXiv:2509.13305* (2025).
>
> > [W2].
>
> We acknowledge the reviewer's concerns regarding reproducibility. The use of human-engineered rubrics for reward modeling is a practical necessity in open-ended domains like travel planning, where ground truth is inherently subjective—a challenge also faced by recent benchmarks (e.g., TP-RAG[1], TripTailor[2]). To enhance reproducibility, we commit to releasing anonymized synthetic queries, simulated sandbox environments, and detailed reward templates in a public repository post-acceptance. While the current implementation relies on DiDi's proprietary APIs for real-world fidelity, our appendix provides comprehensive prompts, data synthesis pipelines, and training configurations to facilitate replication using alternative data sources (e.g., open-source travel APIs).
>
> Reference:
>
> [1] Ni, Hang, et al. "TP-RAG: Benchmarking Retrieval-Augmented Large Language Model Agents for Spatiotemporal-Aware Travel Planning." EMNLP. (2025).
>
> [2] Wang, Kaimin, et al. "Triptailor: A real-world benchmark for personalized travel planning." *Findings of the Association for Computational Linguistics: ACL 2025*. 2025.
>
> > [W3]
>
> We thank the reviewer for raising this point. To address the scale of human evaluation, we have expanded our manual assessment to 200 randomly sampled online user query trajectories (directly collected from the deployed DeepTravel system in DiDi ES App). The collected user query were then used to conduct offline testing of DeepSeek-R1. The results demonstrate that DeepTravel-32B achieves a **human evaluation score of 84.0%**, significantly outperforming DeepSeek-R1 (66.5%).
>
> 200 online user queries|Human Evaluation
> -|-
> DeepTravel-32B|84.00
> DeepSeek-R1|66.50
>
> This confirms the robustness of our framework in real-world settings. We clarify that the system is actively deployed to real users; the offline evaluation uses collected queries to enable comparison with baselines that cannot be directly integrated into the production environment.
>
> > [W4]
>
> All baseline models (including the nine reasoning LLMs) were evaluated under identical conditions: the same system prompt, toolkit, and sandbox environment. No in-context examples or additional fine-tuning were applied to baselines, ensuring a fair comparison. The sole distinction is that DeepTravel models underwent SFT and RL training on our domain-specific data. We note that incorporating in-context examples for travel planning is impractical due to context length constraints—a single multi-turn example can consume ~20K tokens, exceeding typical limits for many LLMs. Thus, our approach of explicit training is both necessary and effective.
>
> > [Q1]
>
> The current sandbox environment supports single-turn training interactions, where the agent processes a full query and generates a complete itinerary in one episode. However, to handle scenarios requiring user clarification or iterative refinement, we synthesized multi-turn dialogue history during training. These dialogue histories serve as input to simulate realistic user–assistant interactions, enabling the agent to continue reasoning effectively within a contextual dialogue setting.
>
> > [Q2]
>
> All baseline LLMs were evaluated with their maximum context window sizes (e.g., 32K tokens for Qwen3-8B, 128K for DeepSeek-R1) to ensure fair comparison. Inference costs were managed by limiting interaction turns to 8 and response lengths to 32K tokens, consistent across all models.

---

> ### Author Response · Authors · 2025-11-25
> **Reminder to Respond to Discussion**
>
> Dear Reviewer SYNB
>
> We sincerely appreciate your comments and suggestions. We have carefully considered them and provided corresponding responses. If you still have any unclear parts of our work, please let us know
>
> Best,
>
> ICLR 2026 Submission 5522 Authors

---

> ### Author Response · Authors · 2025-11-27
> **Looking for your reply**
>
> Dear Reviewer SYNB,
>
> We hope this message finds you well. First and foremost, we would like to sincerely thank you for your thoughtful and constructive feedback on our submission DeepTravel. Your insights, especially regarding the engineering focus and reproducibility concerns, have been incredibly valuable in helping us refine our work. We have carefully considered each of your points in our rebuttal accordingly. Your expertise has pushed us to strengthen both the clarity and the rigor of our contribution. We truly appreciate the time and effort you’ve dedicated to reviewing our paper.
>
> As the decision date approaches, we kindly ask if you might have a moment to review our responses and any updates you may have missed. We would be deeply grateful for your final thoughts.
>
> Thank you again for your guidance and support throughout this process.
>
> Best regards,
>
> Authors

---

### Meta-Review · Area_Chair_wtJf · 2026-01-07

**Summary:**

The paper presents an end-to-end agentic reinforcement learning framework with several carefully designed core modules for autonomous agents. The paper received mixed ratings from four reviewers. Most of them provided initial negative scores and comments regarding the paper's technical contribution and experimental evaluation. The primary concerns include the limited novelty of the method (too engineering, mentioned by two reviewers), the questionable evaluation protocol, the insufficient experimental comparison with state-of-the-art, and the lack of key ablation studies to show the advantages over standard methods. These issues are only partially addressed in the provided rebuttal. To fully address them, a significant modification of the submission is probably needed. Based on all the information from the reviewers and the judgment of the AC, the final recommendation is a rejection of this submission.

**Reviewer Concerns:**

The major concerns from the reviewers include the following aspects: (i) the limited novelty of the method (too engineering, mentioned by two reviewers), (ii) the questionable evaluation protocol, the insufficient experimental comparison with state-of-the-art, and (iii) the lack of key ablation studies to show the advantages over standard methods. To fully address these issues, a large modification of the submission is necessary.

**Reviewer Scores:**

The initial score distribution of this submission is 2, 4, 4, 8. The reviewers did not show intentions to further raise their scores. The rebuttal only partially addressed the raised concerns.

---

### Decision · Program_Chairs · 2026-01-26

Reject